# Softplus Attention with Re-weighting Boosts Length Extrapolation in Large Language Models

**Bo Gao** [* 1 2 3 4]   **Michael W. Spratling** [5]   **Letizia Gionfrida** [4]

## Abstract

Large language models have achieved remarkable success in recent years, primarily due to self-attention. However, traditional Softmax attention suffers from numerical instability and reduced performance as the number of inference tokens increases. This work addresses these issues by proposing a new design principle for attention, viewing it as a two-stage process. The first stage (normalisation) refines standard attention by replacing Softmax with the more numerically stable Softplus followed by $l_1$-normalisation. Furthermore, we introduce a dynamic scale factor based on invariance entropy. We show that this novel attention mechanism outperforms conventional Softmax attention, and state-of-the-art Softmax-free alternatives. Our second proposal is to introduce a second processing stage (sharpening) which consists of a re-weighting mechanism that amplifies significant attentional weights while diminishing weaker ones. This enables the model to concentrate more effectively on relevant tokens, mitigating the attention sink phenomenon, and fundamentally improving length extrapolation. This novel, two-stage, replacement for self-attention is shown to ensure numerical stability and dramatically improve length extrapolation, maintaining a nearly constant validation loss at $16\times$ the training length while achieving superior results on challenging long-context retrieval tasks and downstream benchmarks. Furthermore, symbolic regression experi-

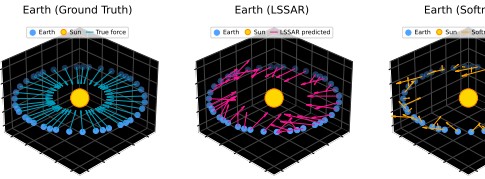

*Figure 1.* Force predictions for Earth's orbit. Left: true forces. Middle: GPT (LSSAR) which mostly captures the radial gravitational structure. Right: GPT (Softmax) predictions are incoherent.

ments demonstrate that our method enables models to recover Newton's gravitational law from orbital trajectory sequences, providing evidence that appropriate attention mechanisms are crucial for foundation models to develop genuine physical world models. Our code is available at `https://github.com/iminfine/freeattn`.

## 1. Introduction

Self-attention has been primarily responsible for the recent success of Large Language Models (LLMs). Self-attention enables models to assess the importance of different words within a sentence, capturing complex relationships and dependencies in the data. Its effectiveness is mainly attributed to the Softmax operation, as when the Softmax operation is omitted or replaced with alternatives, model performance tends to decline (Wortsman et al., 2023; Ramapuram et al., 2024; Shen et al., 2023). Although widely used, Softmax self-attention has two main limitations. Firstly, it suffers from numerical instability due to the exponential function ($e^x$), especially when scaling model sizes to trillions of parameters (Qi et al., 2024). Secondly, as the token length increases during inference, the attention scores calculated by self-attention become smoother and lack distinct peaks. This "attention smoothing" hinders the model's ability to establish connections between relevant tokens, thereby crippling the length extrapolation capabilities of transformers (Chiang & Cholak, 2022; Veličković et al., 2024). This issue is compounded by the "attention sink" phenomenon, where a few initial tokens, often regardless of their semantic importance, attract a disproportionate amount of attention,

[1]Department of Intelligent Manufacturing and Electrical Engineering, Nanyang Normal University, Nanyang 473061, China. [2]Collaborative Innovation Center of Intelligent Explosion-proof Equipment, Henan Province. [3]Baopu Lab. [4]The Department of Informatics, King's College London, Strand, London, WC2R 2LS, UK. [5]Department of Behavioural and Cognitive Sciences, University of Luxembourg, L-4366 Esch-Belval, Luxembourg. Correspondence to: Bo Gao <bo.gao@nynu.edu.cn>.

*Proceedings of the 43rd International Conference on Machine Learning*, Seoul, South Korea. PMLR 306, 2026. Copyright 2026 by the author(s).

leading to suboptimal performance (Xiao et al., 2023). This paper addresses these problems simultaneously by proposing a new design principle for attention mechanisms that offers improved numerical stability and dramatically better performance at large token lengths. Our core proposal is a new attention architecture consisting of two stages: a **Normalisation Stage** followed by a **Sharpening Stage**.

To develop a better normalisation stage, we first analysed the standard Softmax function. We deconstructed it into its two functional components: a non-linear positivity transformation ($e^x$) and a subsequent scaling by the $l_1$-norm. Our experiments revealed that the $l_1$-norm scaling is the critical component for maintaining model performance. This insight frees us to redesign the attention process for better stability and performance. We replace the exponential function with the more numerically stable Softplus activation and incorporate a dynamic length scale factor based on invariance entropy. This creates our new normalisation stage, a novel mechanism we call *Length Scaled Softplus Attention* (LSSA), which outperforms the standard attention mechanism not only at the training sequence length but also for longer sequences.

However, a normalisation stage alone does not solve the attention smoothing problem. Therefore, we introduce a re-weighting mechanism that sharpens the attention distribution. Applied after LSSA, this mechanism amplifies critical token relationships and suppresses noise using a power transformation. This inherently sharpens attention peaks without needing post-hoc fixes like positional interpolation (Chen et al., 2023; Li et al., 2023), which retroactively stretches embeddings but fails to address the root cause of attention smoothing.

The combination of these two proposed mechanisms, LSSA (for stable normalisation) and re-weighting (for sharpening), results in our final model, LSSAR. The effectiveness of this two-stage design is starkly demonstrated in challenging "needle-in-a-haystack" passkey retrieval tasks, where LSSAR succeeds far beyond its training length while standard attention fails completely. Furthermore, LSSAR effectively mitigates the attention sink phenomenon by promoting a more balanced attention distribution. LSSAR maintains a nearly constant validation loss even at $16\times$ the training token length and translates its superior internal metrics into better performance on real-world NLP tasks.

Beyond standard benchmarks, we evaluate whether LSSAR helps models capture physically meaningful structure through symbolic regression experiments on planetary orbits. As illustrated in Fig. 1, a GPT-109M model equipped with LSSAR predicts force vectors that are more consistent with the Newtonian radial direction than the standard Softmax baseline. Symbolic regression identifies an inverse-square form ($F \propto m/r^2$) from LSSAR's outputs, whereas

standard attention fails to match this physical structure. Notably, even state-of-the-art LLMs that are assumed to be substantially larger than our GPT-109M model (o3, Claude 4 Sonnet, and Gemini 2.5 Pro) do not recover the inverse-square relation in the same protocol (Vafa et al., 2025), suggesting that appropriate attention mechanisms are crucial for foundation models to develop physically grounded world-model representations.

In summary, this paper makes several contributions:

1. We introduce a novel Softmax-free attention mechanism, LSSA, which incorporates a dynamic length scale factor based on invariance entropy, demonstrating superior performance compared to standard attention mechanisms across a variety of sequence lengths.

2. We propose a novel architectural re-weighting mechanism that fundamentally redesigns how attention scores are computed, inherently sharpening token relevance by amplifying critical weights and suppressing noise.

3. When LSSA is combined with the proposed re-weighting method, the resulting model (LSSAR) not only excels in length extrapolation while ensuring numerical stability, as demonstrated by its success in passkey retrieval tasks where standard attention fails, but also translates into superior performance in real-world applications.

4. We demonstrate through symbolic regression that LSSAR induces inductive biases aligned with physically meaningful sequence structure. Unlike standard Softmax attention, which fails to recover a compact physical relation in this setup, LSSAR recovers an inverse-square functional dependence from orbital trajectories.

## 2. Method

The proposed attention mechanism consists of two stages. The first involves normalising the raw attention scores to create a stable distribution, and is described in Sect. 2.1. The second re-weights this distribution to sharpen focus and enhance length extrapolation, as described in Sect. 2.2.

### 2.1. The Normalisation Stage

#### 2.1.1. SOFTMAX DECOMPOSITION

$$\mathbf{A} = \text{Softmax}\left(\frac{\mathbf{Q}\mathbf{K}^T}{\sqrt{d}} + \mathbf{M}\right) \quad (1)$$

Scaled dot-product attention transforms queries ($\mathbf{Q}$), keys ($\mathbf{K}$), and values ($\mathbf{V}$) into an output. First, attention scores $\mathbf{A}$ are produced via Eq. (1). Here, $\mathbf{Q}, \mathbf{K}, \mathbf{V} \in \mathbb{R}^{L \times d}$, where $L$ is the sequence length and $d$ is the dimensionality. The

optional term $\mathbf{M}$ is a mask matrix of shape $L \times L$, which is essential in causal self-attention. The mask $\mathbf{M}$ is constructed with zeros on and below the diagonal and $-\infty$ above it, ensuring the attention mechanism only considers past and present tokens. The attention scores are then used to compute the output as $\mathbf{AV}$.

The Softmax function is a cornerstone of the attention mechanism. Its non-negative outputs are often viewed as an important factor for stable attention. However, prior results with other non-negative activation functions suggest that non-negativity alone is not sufficient for good performance (Wortsman et al., 2023; Ramapuram et al., 2024; Shen et al., 2023).

To investigate this decomposition, we conducted experiments that modify the sign and normalisation of attention scores. Our results (see Tab. A.4 in Appendix B.1.1) suggest that Softmax's effectiveness comes from the combination of a nonlinear positivity transformation and subsequent $l_1$ normalisation. Within this decomposition, the normalisation step plays a central role in preserving performance, rather than positivity alone.

Motivated by this observation, we decompose the Softmax operation into a non-linear positivity transformation, $\phi(\mathbf{x}) = e^{\mathbf{x}}$, followed by an $l_1$-norm scaling:

$$\text{Softmax}(\mathbf{x}) = \frac{e^{\mathbf{x}}}{\sum_j e^{x_j}} = \frac{e^{\mathbf{x}}}{\|e^{\mathbf{x}}\|_1} = \frac{\phi(\mathbf{x})}{\|\phi(\mathbf{x})\|_1} \qquad (2)$$

This allows us to generalise attention as shown in Eq. (3).

$$\mathbf{A} = \phi\left(\frac{\mathbf{QK}^T}{\sqrt{d}}\right) \otimes \mathbf{M}'$$
$$\mathbf{A}_i \leftarrow \frac{\mathbf{A}_i}{\|\mathbf{A}_i\|_1} \qquad (3)$$

The subscripts $i$ and $j$ represent the row and column indices respectively. $\otimes$ denotes element-wise multiplication. The mask $\mathbf{M}'$ is constructed with ones on and below the diagonal and zeros above it. It is evident that the original attention function (Eq. (1)) is a special case of this general form where $\phi(\mathbf{x}) = e^{\mathbf{x}}$.

### 2.1.2. LENGTH SCALED SOFTPLUS ATTENTION

Based on the general form of Eq. (3), we now introduce *Length Scaled Softplus Attention* (LSSA), a novel attention variant designed to enhance the scalability and performance of LLMs. LSSA replaces the exponential function with $\text{Softplus}(x) = \log(1 + e^x)$ to introduce non-linearity while maintaining smooth gradients essential for stable training dynamics. Empirical testing of several widely-used activation functions in place of $\phi$ in Eq. (3), revealed that Softplus delivers the best performance (see Tab. A.6), justifying its adoption. As described below, LSSA also introduces a novel

scaling factor that accounts for both the sequence length and the model's dimensionality, thereby addressing limitations associated with traditional attention methods in handling long sequences during inference.

In contrast to scaled dot-product attention, cosine similarity attention employs the $l_2$-norm for each row of $\mathbf{Q}$ and $\mathbf{K}$. This approach has been shown to produce more moderate attention weights, which can enhance performance across various tasks (Henry et al., 2020; Dehghani et al., 2023; Liu et al., 2022). Furthermore, the $l_2$-norm restricts the dot product values to the interval $[-1, 1]$, effectively placing them in the region where the derivative of the Softplus function, the Sigmoid $\sigma(x) = 1/(1 + e^{-x})$, exhibits a steep slope. This characteristic ensures that the gradients remain distinct across different inputs, rendering the attention mechanism acutely sensitive to small variations in the latent representations.

However, in high-dimensional spaces, as the number of dimensions increases, two randomly selected vectors are likely to become orthogonal. This phenomenon causes the elements of the product of $\mathbf{Q}_i/\|\mathbf{Q}_i\|_2$ and $\mathbf{K}_i^T/\|\mathbf{K}_i\|_2^T$ to approach zero, thereby compressing the dynamic range of the attention scores. Consequently, a scale factor becomes essential for cosine similarity attention to restore discriminability, and the factor should be associated with the dimensionality $d$.

Previous work demonstrated that replacing the traditional scaling factor $1/\sqrt{d}$ with $\log L/\sqrt{d}$ in scaled dot-product attention enhances the length extrapolation capabilities of transformers (Chiang & Cholak, 2022; Nakanishi, 2025). Furthermore, Su (2021) highlighted that the inclusion of the $\log L$ factor aids in maintaining entropy invariance with different token length, thereby facilitating better extrapolation to unknown sequence lengths. We extend this concept by introducing a dynamic length scale factor that adapts to the varying number of attended tokens in each row of the attention matrix. Specifically, we set the scaling factor to $\log d \log \mathbf{N}$, where $\mathbf{N}$ is an $L \times L$ matrix where each element in row $i$ is equal to $i$ (the number of tokens attended to in that row). This ensures the attention mechanism remains robust across varying sequence lengths. The formulation of LSSA is mathematically defined in Eq. (4).

$$\mathbf{Q}_i \leftarrow \frac{\mathbf{Q}_i}{\|\mathbf{Q}_i\|_2}, \mathbf{K}_i \leftarrow \frac{\mathbf{K}_i}{\|\mathbf{K}_i\|_2}$$
$$\mathbf{A} = \text{Softplus}\left((\log d \log \mathbf{N}) \otimes \mathbf{QK}^T\right) \otimes \mathbf{M}' \qquad (4)$$
$$\mathbf{A}_i \leftarrow \frac{\mathbf{A}_i}{\|\mathbf{A}_i\|_1}$$

It is important to note that due to hardware limitations, the proposed LSSA was evaluated with a sequence length of $L = 1024$ and dimension $d = 64$. For LLMs trained with longer sequence lengths and higher dimensionalities, the

scaling factor may require further adjustment. However, the component $\log \mathbf{N}$ should remain unchanged, as it is crucial to maintain the length extrapolation capability of the model.

## 2.2. The Sharpening Stage

### 2.2.1. RATIONALE FOR TWO-STAGE ATTENTION

While the normalisation stage provides a stable attention distribution, it fundamentally retains the dense nature of Softmax, activating all input tokens regardless of their relevance. This counter-intuitive behaviour exacerbates the attention smoothing problem (Chiang & Cholak, 2022; Veličković et al., 2024), where attention scores lack distinct peaks as the sequence length increases during inference, hindering the model's ability to distinguish signal from noise.

A naive solution is to replace $\phi$ in Eq. (3) with a sparsity-inducing activation function, such as ReLU. However, as shown in Tab. A.6, ReLU-based attention results in performance degradation. In ReLU-based attention, positive values are predominantly concentrated around the diagonal, while off-diagonal positions often remain negative and are suppressed to zero. This creates a 'dead neuron' phenomenon during training: once a token's score falls below zero, it loses all gradient feedback, effectively isolating it from the optimisation process and restricting the model's capacity to capture long-range dependencies.

Fundamentally, the efficacy of the attention mechanism stems from the synergy between two distinct functional components: a consistent non-linear transformation and a subsequent normalisation. Functions such as Softplus and $e^x$ transform raw scores while keeping every token's contribution non-zero. Unlike ReLU-based mappings, they do not collapse any admissible input to exactly zero before normalisation. This property ensures that every token remains eligible to participate in the subsequent $l_1$-norm competition. The $l_1$-norm then serves two critical roles that are essential for stable training. In the forward pass, it functions as a *weighted sum* operator (ensuring $\sum |A_{ij}| = 1$), which maintains the scale stability of the output representations regardless of the input magnitude or sequence length. In the backward pass, the $l_1$-norm introduces a mechanism of *lateral inhibition* via gradient coupling (see biological justification in Appendix A.2). By linking all tokens through the denominator term, the optimisation of 'winner' tokens propagates gradients back to suppress 'losers'. This global coupling guarantees that even currently suppressed tokens receive gradient feedback, thereby preventing the dead neuron issue associated with hard thresholding.

### 2.2.2. ATTENTION RE-WEIGHTING

The theoretical insight reframes both standard Softmax and our LSSA as a unified **Normalisation Stage**, which estab-

lishes a numerically stable and gradient-connected landscape. It is upon this robust foundation that we introduce our re-weighting mechanism, defined in Eq. (5), as a distinct **Sharpening Stage**. By applying the shift and re-weighting operations *after* the gradient pathways have been secured by the normalisation stage, the proposed method achieves the desired sparsity and noise filtering without sacrificing trainability.

$$
\begin{aligned}
\mathbf{A} &\leftarrow \text{ReLU}^p \left( \mathbf{A} \otimes \mathbf{N} - \mathbf{O} \right) \\
\mathbf{A}_i &\leftarrow \frac{\mathbf{A}_i}{\|\mathbf{A}_i\|_1}
\end{aligned}
\tag{5}
$$

Here, the normalised scores $\mathbf{A}$ are first scaled by the token count $\mathbf{N}$ and then shifted by a matrix $\mathbf{O}$ (an offset matrix of ones, with zeros in the first three rows to prevent instability). This centres the distribution around zero. The $\text{ReLU}^p$ function then masks scores below zero and sharpens the remaining positive scores by raising them to the power of a hyper-parameter $p$. A final scaling by the $l_1$-norm ensures the output is a valid probability distribution. The re-weighting mechanism is an additional component of our proposed attention layer, we refer to the combination of LSSA (stage 1) and this re-weighting mechanism (stage 2) as LSSAR in the rest of this paper. The complete LSSAR algorithm is provided in Algorithm 1 in Appendix A.1.

This power operation is key to solving the attention smoothing problem. Let $x_1, x_2, \ldots, x_n$ be the positive elements in one row of the ReLU output from Eq. (5), and let $M = \max_{1 \le k \le n} x_k$. The re-weighted value of the $j$-th entry is

$$
\overline{x}_j = \frac{x_j^p}{\sum_{k=1}^n x_k^p}.
\tag{6}
$$

If $x_j < M$, then

$$
\overline{x}_j = \frac{(x_j/M)^p}{\sum_{k=1}^n (x_k/M)^p} \to 0 \quad \text{as } p \to \infty.
\tag{7}
$$

If $t$ entries attain the maximum value $M$, then each of those entries converges to $1/t$ as $p \to \infty$. In the generic case where the maximum is unique, this reduces to a one-hot limit on the largest entry. The tied-maximum case is a boundary case, in which the limiting mass is shared uniformly among the tied maxima. This limit shows that increasing $p$ suppresses smaller values while concentrating mass on the largest surviving entries after the ReLU shift. This property ensures that the attention mechanism maintains sharp, distinct peaks even with longer sequences, thereby preserving the model's ability to focus on the most relevant tokens. This offers distinct advantages over projection-based sparse mechanisms (e.g., Sparsemax (Martins & Astudillo, 2016)), particularly in terms of computational efficiency and hardware compatibility, as discussed in Appendix A.3. From a

thermodynamic perspective (see Appendix A.4), this process can be viewed as active entropy minimisation, with $p$ acting as an inverse temperature coefficient. Furthermore, geometric analysis (Appendix A.4) suggests that LSSAR prevents 'manifold drift' by anchoring representation updates to a local semantic neighbourhood.

# 3. Experiments

The experiments were conducted using 8 NVIDIA A100 80GB GPUs. We utilised the GPT-2 small architecture (124 million parameters; Radford et al., 2019), replacing the original absolute position embeddings with Rotary Position Embeddings (RoPE; Su et al., 2024).

All models were trained using a sequence length of 1024 on the FineWeb-10B dataset (Penedo et al., 2024), which consists of 10.2 billion training tokens distributed across 18,865 training steps, along with 0.1 billion validation tokens. Full details of the optimisation setup, batch construction, and numerical precision are provided in Appendix B.

Due to page limitations, we provide extensive supplementary experiments and analysis in Appendix B. These include: (1) a decomposition study of the Softmax function (Tab. A.5), which empirically confirms the central role of $l_1$-normalisation; (2) a comparison of different activation functions (Tab. A.6), demonstrating that Softplus performs best among the tested alternatives; (3) additional LSSAR+PI experiments in Appendix B.2, showing that LSSAR can be combined with position interpolation; (4) an analysis of the re-weighting parameter $p$ in Appendix B.1.4 and an adaptive position-dependent $p_i$ schedule in Appendix B.3; and (5) computational analysis and scaling experiments with filtered data in Appendix B.4 and Appendix B.6, respectively.

## 3.1. Comparison with Softmax-Free Attention Methods

We compared the proposed LSSA and LSSAR with leading Softmax-free methods using the same GPT-2-124M model. These attention functions include two that use Sigmoid (one proposed in Ramapuram et al. (2024) and the other produced when using Sigmoid for $\phi$ in Eq. (3) as used in Tab. A.6), and three ReLU-based attention methods from (Wortsman et al., 2023; Li et al., 2022; Shen et al., 2023). The original Sigmoid attention (Ramapuram et al., 2024) computes attention scores as Sigmoid$(x - b)$, where the bias term $b = -\log L$ shifts the input distribution to prevent saturation. We evaluated two additional variants. The first variant alters the hyperparameter $b$ from $-\log L$ to $-\log \mathbf{N}$. The second variant maintains this modification while subsequently applying the $l_1$-norm to each row of the attention matrix, ensuring each row sums to one. We additionally include Scalable-Softmax (SSMax) (Nakanishi, 2025) as a direct baseline for methods that modify the effective atten-

tion temperature. SSMax is trained under the same setting, with its learnable scaling parameter initialised as $s = 0.43$. The experimental findings are detailed in Tab. 1.

At an inference sequence length of 1K, all attention variants perform similarly, with only slight differences in validation loss. Specifically, SSMax achieves the lowest validation loss, while Sigmoid (Ramapuram et al., 2024)($b = -\log \mathbf{N}, l_1$-norm) and LSSA also perform slightly better than the standard Softmax attention. As sequence length increases, performance differences become more evident. The standard Softmax attention experiences performance decline with longer sequences but remains relatively stable compared to most other methods.

Sigmoid attention (Ramapuram et al., 2024) shows significant performance decline at longer sequences, with validation loss rising sharply at 8K tokens. Theoretically, this method employs a shifted activation Sigmoid$(x - b)$, where the bias term $b$ effectively shifts the input distribution leftwards. This is designed to prevent the inputs from falling into the right-hand saturation region of the Sigmoid function, thereby maintaining sensitivity to large scores. However, our results indicate that this element-wise shift alone is insufficient. The modified Sigmoid (Ramapuram et al., 2024)($b = -\log \mathbf{N}$) shows some improvement, suggesting that the proposed hyperparameter $\log \mathbf{N}$ is more effective than $\log L$, yet the model still faces challenges with longer sequences. Crucially, incorporating the $l_1$-norm in Sigmoid (Ramapuram et al., 2024)($b = -\log \mathbf{N}, l_1$-norm) dramatically recovers performance, aligning it closely with standard Softmax attention, underscoring the importance of $l_1$-norm in attention mechanisms.

ReLU-based attention variants (Wortsman et al., 2023; Li et al., 2022; Shen et al., 2023) also experience notable performance decline with longer sequences, though not as drastically as the basic Sigmoid attention. This indicates that while ReLU-based methods may be effective for shorter sequences, they might not be ideal for managing longer-range dependencies. We conjecture that this degradation may be attributable to the 'dead neuron' phenomenon: as sequence length increases, a larger proportion of off-diagonal attention scores fall below zero and are suppressed, potentially severing gradient pathways for distant tokens. A theoretical analysis of this mechanism is provided in Sect. 2.2.1.

To demonstrate the effectiveness of the proposed re-weighting mechanism, the attention scores generated by Softmax, Sigmoid (Ramapuram et al., 2024)($b = -\log \mathbf{N}, l_1$-norm), and LSSA were re-weighted with $p = 3$ and $p = 15$. As shown in Tab. 1, the length extrapolation ability provided by the re-weighting can improve the performance of all the tested attention mechanisms. For Softmax, performance was improved slightly using $p = 3$, but was harmed when using $p = 15$. For the other attention mech-

*Table 1.* Comparison of Softmax-attention and state-of-the-art Softmax-free attention mechanisms across different sequence lengths. Bold font indicates the best result for each sequence length.

| Attention Mechanism | 1K | 2K | 4K | 8K |
|---|---|---|---|---|
| *Base attention mechanisms (no re-weighting)* | | | | |
| Softmax | 3.1911 | 4.1662 | 5.4513 | 6.2823 |
| SSMax (Nakanishi, 2025) | **3.1705** | 4.0915 | 5.4606 | 6.4384 |
| Sigmoid (Ramapuram et al., 2024) | 3.1935 | 7.4554 | 11.8355 | 14.4995 |
| Sigmoid (Ramapuram et al., 2024)$(b = -\log \mathbf{N})$ | 3.1930 | 6.5830 | 8.4679 | 9.5939 |
| Sigmoid (Ramapuram et al., 2024)$(b = -\log \mathbf{N}, l_1\text{-norm})$ | 3.1849 | 4.3470 | 5.5544 | 6.1846 |
| Sigmoid (Tab. A.6) | 3.2000 | 4.3811 | 5.8465 | 6.5701 |
| ReLU (Wortsman et al., 2023) | 3.2143 | 6.2662 | 8.4982 | 10.3460 |
| ReLU (Li et al., 2022) | 3.2006 | 4.5192 | 5.6924 | 6.4561 |
| ReLU (Shen et al., 2023) | 3.2155 | 6.5573 | 8.9072 | 10.7266 |
| LSSA | 3.1905 | 4.1301 | 5.2960 | 5.9403 |
| *Re-weighted variants ($p = 3$)* | | | | |
| Softmax | 3.1879 | 4.0277 | 5.2842 | 6.2339 |
| Sigmoid (Ramapuram et al., 2024)$(b = -\log \mathbf{N}, l_1\text{-norm})$ | 3.1841 | 3.7387 | 4.9286 | 5.7450 |
| LSSAR | 3.1782 | 4.2383 | 5.4056 | 6.3007 |
| *Re-weighted variants ($p = 15$)* | | | | |
| Softmax | 5.3878 | 5.9491 | 6.5276 | 7.0183 |
| Sigmoid (Ramapuram et al., 2024)$(b = -\log \mathbf{N}, l_1\text{-norm})$ | 3.2171 | 3.3499 | 3.6108 | 3.8587 |
| LSSAR | 3.1905 | **3.1930** | **3.2291** | **3.3171** |

anisms, $p = 3$ was most effective for sequence lengths of 1K, while $p = 15$ was best for the longer sequences. For $p = 15$, LSSAR achieves the best long-context performance among the tested variants, with validation loss remaining comparatively flat up to 8K tokens. The gradient analysis in Appendix A.6 explains this contrast: under stronger sharpening, Softmax exhibits exponential gradient decay in the high-confidence regime, whereas LSSAR exhibits polynomial gradient decay and therefore retains a more effective optimisation signal. This quantitative robustness is further corroborated by the qualitative attention map visualisations in Appendix B.5 (see Fig. A.1), where LSSAR displays a more balanced distribution across tokens with sharper, non-collapsing peaks and markedly reduced attention sink compared to standard Softmax, confirming its enhanced ability to preserve salient long-range dependencies.

The above experiments use standard RoPE without modifying the positional encoding. Since attention re-weighting and positional interpolation operate on different parts of the Transformer, they can also be combined. We therefore include additional LSSAR+PI experiments in Appendix B.2, showing that LSSAR remains effective under continued long-context finetuning with position interpolation.

### 3.2. Long Context Passkey Retrieval

To offer a more direct and challenging evaluation of length extrapolation, we employed the long-context passkey retrieval task (Mohtashami & Jaggi, 2023). This needle-in-a-haystack test is specifically designed to assess a model's ability to identify a single, crucial, piece of information

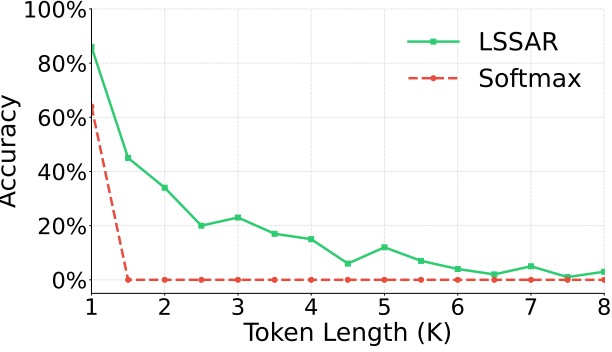

*Figure 2.* Passkey retrieval accuracy for LSSAR ($p = 15$) and standard Softmax attention. Accuracy is averaged over 100 trials with the passkey placed at random positions within the sequence.

within a long and distracting context.

We compared LSSAR ($p = 15$) against the standard Softmax attention baseline. The results, shown in Fig. 2, reveal a critical failure in the baseline model. While Softmax attention achieves 64% accuracy within its training length, its performance catastrophically collapses to 0% for all tested lengths of 1.5K tokens and beyond. This demonstrates its inability to overcome attention smoothing, effectively losing the passkey as the context grows. In stark contrast, LSSAR not only achieves a much higher accuracy of 86% at the 1K training length but also demonstrates remarkable robustness in extrapolation. Its accuracy degrades gracefully from 45% at 1.5K tokens, 20% at 4K tokens, and it maintains non-zero accuracy even at 8K tokens. This sustained, non-zero performance provides clear evidence that the re-weighting

*Table 2.* Zero-shot performance of the models with Softmax attention and LSSAR on downstream tasks. The best scores in each column indicated in bold.

| | ARC-E | ARC-C | HellaSwag | PIQA | MMLU | SciQ | SummScreen |
|---|---|---|---|---|---|---|---|
| Softmax | 39.77 | **23.72** | 32.42 | 64.09 | 22.97 | 60.6 | 1.682 |
| LSSAR ($p = 15$) | **40.57** | 22.61 | **33.03** | **65.34** | 22.97 | **62.1** | **6.309** |

*Table 3.* Symbolic regression results comparing different models. Since the solar mass is fixed in this setting, the fitted constant corresponds to $Gm_{\text{sun}}$.

| Model | Recovered Function | Fitted Constant |
|---|---|---|
| Oracle (KNN) | $F \propto \frac{m_1}{r^2}$ | 25.85 |
| GPT (Softmax) | $F \propto \frac{\left(\sin\left(\frac{1}{\sin(r-0.24)}\right)+1.45\right)}{\frac{1}{r}+m_2}$ | N/A |
| GPT (LSSAR) | $F \propto \frac{m_1}{r^2}$ | 27.02 |
| Ground Truth | $F \propto \frac{m_1}{r^2}$ | $4\pi^2 \approx 39.48$ |

mechanism successfully sharpens the model's focus, allowing it to pinpoint the relevant information even in sequences far exceeding its training context and overcoming a critical limitation of standard Softmax attention. Complementary passkey results for LSSAR combined with Position Interpolation are reported in Tab. A.8 in Appendix B.2. Additional experiments on passkey retrieval with different model scales and filtered FineWeb-Edu data are provided in Fig. A.3 in Appendix B.6.

### 3.3. Downstream Evaluation

To demonstrate the practical efficacy of LSSAR, we evaluate its zero-shot performance against Softmax attention on six standard benchmarks: ARC (Clark et al., 2018), HellaSwag (Zellers et al., 2019), PIQA (Bisk et al., 2020), MMLU (Hendrycks et al., 2020), SciQ (Welbl et al., 2017), and SummScreen (Chen et al., 2022), using the `lm-evaluation-harness` framework (Gao et al., 2024). For ARC-E, ARC-C, HellaSwag, PIQA, and SciQ, we report normalised accuracy; for MMLU, we use standard accuracy; and for SummScreen, we adopt the ROUGE-1 score to measure summarisation quality.

LSSAR outperforms Softmax attention in five of these tasks, and has performance competitive with that of Softmax attention in the other two (Tab. 2). Notably, LSSAR excels on the long-context summarisation benchmark, SummScreen, achieving a nearly fourfold improvement score over Softmax attention. This highlights LSSAR's ability to synthesize information over sequences of up to 2K tokens, aligning with its design for robust length extrapolation. Further downstream evaluation results on different model scales can be found in Appendix B.6. Additional analysis on the

LAMBADA benchmark, which tests long-range contextual understanding, is provided in Appendix B.7.

### 3.4. Symbolic Regression on Physical Structure

To evaluate whether LSSAR helps models capture physically meaningful structure, we conducted symbolic regression experiments following the force-prediction protocol of Vafa et al. (2025). We trained a 109M parameter GPT-2 model on 10M simulated orbital trajectory sequences, replacing the standard Softmax attention with our proposed LSSAR mechanism, while keeping all other architectural and training hyperparameters unchanged. The model was pretrained to predict planetary positions and subsequently fine-tuned on a force prediction task, where the output was the gravitational force magnitude $F$ given the planetary state (mass $m_1$, solar mass $m_2$, and distance $r$).

Following fine-tuning, PySR (Cranmer, 2023) was used to perform symbolic regression on the model-predicted force values, rather than on ground-truth equations, searching for interpretable mathematical expressions that best fit the model's predictions. The true gravitational law follows Newton's formula:

$$F = G \cdot \frac{m_1 \cdot m_2}{r^2} \tag{8}$$

where $m_2 = m_{\text{sun}}$ is fixed in the solar-system setting, so $G \cdot m_2 = G \cdot m_{\text{sun}} = 4\pi^2 \approx 39.48$ in astronomical units.

As shown in Fig. 1, LSSAR generates force vectors that are more consistent with the Newtonian directionality than the Softmax baseline. Symbolic regression identifies an inverse-square form, $F = 27.02 \times m_1/r^2$, which captures the distance dependence and linear mass dependence. Recovering this inverse-square structure is the central evidence that LSSAR helps the model infer physically meaningful relations from orbital trajectories. In contrast, standard GPT fails to match the physical structure, producing nonsensical equations (Vafa et al., 2025).

As shown in Tab. 3, LSSAR's fitted constant (27.02) is closer to the theoretical value ($4\pi^2 \approx 39.48$) than the Oracle (KNN) baseline with direct access to true force labels (25.85). The fitted constant still differs from the theoretical value by approximately 31.6%, which may be affected by data noise, optimisation randomness, model capacity, and the limited force-prediction finetuning setup. We therefore view the coefficient mismatch as a limitation of numerical

calibration, while the recovered inverse-square dependence remains the key evidence of physical-structure recovery. The Oracle baseline learns to interpolate from true gravitational force labels, whereas LSSAR observes orbital trajectory sequences during pretraining and is fine-tuned on force prediction. This comparison suggests that the LSSAR attention mechanism can provide a useful inductive bias for recovering compact physical structure.

To further contextualise these results, Vafa et al. (2025) also tested state-of-the-art large language models, including o3, Claude 4 Sonnet, and Gemini 2.5 Pro, on the same force-prediction task using in-context learning. Those models did not recover the inverse-square relation, producing trivial expressions such as $F \propto m_1$ or $F \propto 1/(m_2 - 0.50)$.

Overall, the LSSAR-based model yields an interpretable inverse-square form, whereas the Softmax baseline and the in-context LLM baselines fail to match this physical structure. These results indicate that LSSAR's attention mechanism induces inductive biases aligned with compact physical structure in this controlled setting. A mechanistic interpretation of why LSSAR may improve symbolic-regression outcomes is provided in Appendix A.2.1. Visualisations of force predictions for other planets in the solar system are provided in Appendix B.8.

## 4. Discussion

Due to limited computing resources, our main experiments evaluate LSSAR on GPT-2-124M models trained at a sequence length of 1024 and tested up to 16K tokens. The results in Fig. A.3 show that LSSAR maintains a relatively stable validation loss over this extrapolation range, suggesting that the proposed attention mechanism improves the intrinsic length-generalisation behaviour of the Transformer. Additional experiments on 45M and 355M models are provided in Appendix B.6, where LSSAR shows consistent improvements over the Softmax baseline across the tested model sizes. Furthermore, we discuss implications for reasoning models in Appendix B.7.1.

While our PyTorch implementation incurs substantial computational overhead, this should be treated as a current implementation constraint rather than a settled efficiency result. In the current implementation, LSSAR is noticeably slower at inference and uses substantially more memory during training than standard attention. Our analysis in Appendix B.4 suggests that LSSAR may be amenable to future fused implementations, but we do not yet provide an optimised kernel or an efficiency result comparable to standard high-performance attention implementations.

## 5. Limitations

We emphasise several limitations of the present study. First, the empirical evidence currently comes from models in the 45M to 355M parameter range and from training runs that are much shorter than those used to train production-scale language models. The present results should therefore be interpreted as mechanism-level evidence rather than as definitive large-scale validation.

Second, the principal extrapolation experiments train at sequence length 1024 and evaluate zero-shot at 2K, 4K, 8K, and 16K. This setting is useful for isolating intrinsic length generalisation, but it does not correspond to a production training pipeline in which long-context examples are gradually introduced during training or in which the model is further adapted at the target context length. The reported zero-shot results should thus be read as controlled extrapolation measurements rather than as a complete long-context recipe.

Third, LSSAR remains an exact $O(L^2)$ attention mechanism rather than a linear-attention or state-space alternative. Its present implementation is therefore intended to test the attention mechanism itself, not to provide a drop-in efficiency improvement over highly optimised attention kernels.

Fourth, the stability of LSSAR relative to standard Softmax attention in much deeper networks remains untested, especially because Softplus and its derivative can saturate in certain regimes. Larger-scale validation, deeper-network analysis, and kernel-level optimisation therefore remain important directions for future work.

## 6. Related Work

**Softmax-free Attention.** Previous research has investigated Softmax-free attention by substituting the Softmax function with the ReLU activation (Li et al., 2022; Shen et al., 2023; Wortsman et al., 2023; Hron et al., 2020; Bai et al., 2023; Fu et al., 2023), SquaredReLU activation (Hua et al., 2022), and Sigmoid activation (Ramapuram et al., 2024), as well as examining purely linear attention (Qin et al., 2022; Tsai et al., 2019; Katharopoulos et al., 2020; Han et al., 2023; Arora et al., 2024; Lu et al., 2021). However, none of these approaches outperform the original Softmax attention. In contrast, the proposed LSSA demonstrates enhanced numerical stability and superior performance across various token lengths.

It is also important to distinguish our work from two other lines of research for long-context modeling. The first improves computational efficiency by approximating the attention matrix, leading to linear attention mechanisms such as Performer (Choromanski et al., 2020). The second line of research proposes entirely new architectures, such as intro-

ducing sparse attention patterns as in LongFormer (Beltagy et al., 2020), or replacing the attention paradigm altogether with State Space Models like Mamba (Gu & Dao, 2024). In contrast to these approaches, LSSAR remains within the exact, quadratic-time, attention paradigm. Our contribution is not a new architecture or an approximation, but a direct, drop-in replacement for the standard attention mechanism designed to fundamentally improve its numerical stability and length extrapolation capabilities within the conventional Transformer framework.

Among these linear-attention variants, FlattenTransformer is particularly relevant because it also uses amplification to focus attention weights (Han et al., 2023). The mechanism, however, is different: FlattenTransformer amplifies query-key features before applying a linear attention approximation, whereas LSSAR sharpens the exact pairwise attention scores after the first-stage mapping and row-wise normalization. This distinction motivates our focus on attention re-weighting in the exact attention space.

**Attention Re-weighting.** The non-linear re-weighting mechanism introduced by softmax attention ($l_1$-normalization) has been shown to concentrate the distribution of attention weights, thereby stabilising the training process (Titsias, 2016; Gao & Pavel, 2017; Jang et al., 2016; Qin et al., 2022). Our empirical findings further demonstrate its essential role in maintaining the performance of LLMs. Moreover, we introduce a novel perspective: a non-linear positivity transformation followed by $l_1$-normalization. Inspired by the classic normalization-ReLU structure, the proposed re-weighting mechanism masks less relevant tokens and amplifies the relevant ones, which boosts the length extrapolation ability of underlying models. This provides the deep learning community with a deeper understanding of the attention mechanism within transformers.

**Length Extrapolation.** Positional embeddings (Su et al., 2024; Chen et al., 2023; Chi et al., 2022; Kiyono et al., 2021; Golovneva et al., 2024; He et al., 2024; Huang et al., 2020; Li et al., 2023; Likhomanenko et al., 2021; Liu et al., 2023; Wang et al., 2024; Zheng et al., 2024) play a vital role in transformer architectures by providing essential information about the positions of tokens within a sequence, which is considered a key factor in enhancing length extrapolation (Kazemnejad et al., 2023). Among these embeddings, RoPE is particularly noteworthy; it forms the foundation of many modern LLMs, including GPT-4 (Achiam et al., 2023), Llama3 (Dubey et al., 2024), Deepseek-v3 (Liu et al., 2024), DeepSeek-R1 (Guo et al., 2025) and Qwen3 (Yang et al., 2025). Unlike RoPE-based extrapolation techniques that compensate for sequence length limitations through positional embedding adjustments (Chen et al., 2023; Li et al., 2023; kaiokendev, 2023; bloc97, 2023b;a; emozilla,

2023), LSSAR redefines the core attention mechanism itself. The re-weighting operation is not merely an auxiliary technique but a structural enhancement to the attention computation, ensuring sharp peaks and stable gradients by design. This indicates that a well-designed attention mechanism like LSSAR is essential for RoPE-based LLMs to enhance their length extrapolation, suggesting that LSSAR may serve as a general strategy to bolster the length extrapolation abilities of most contemporary LLMs.

## 7. Conclusion

This paper introduced two novel improvements to attention mechanisms. The first normalizes scores using LSSA, a mechanism built on the Softplus function and a dynamic length scale factor, which outperforms standard Softmax. The second applies a unique re-weighting mechanism that sharpens the attention distribution. The combined model, LSSAR, demonstrates a remarkable ability to extrapolate to longer sequences while maintaining numerical stability and mitigating the attention sink phenomenon. Downstream evaluations confirm that these architectural improvements translate into superior performance on a range of tasks.

Beyond conventional benchmarks, our symbolic regression experiments reveal a deeper insight: LSSAR enables models to recover the inverse-square structure of Newtonian gravity from orbital trajectories, while standard Softmax attention fails to match this physical structure. This demonstrates that appropriate attention mechanisms are crucial for foundation models to develop physically grounded world models.

## Acknowledgements

We acknowledge financial support from the Special Project of Nanyang Normal University (No. 2024ZX033) and the Natural Science Foundation of Henan (Nos. 252300420979 and 262300422563).

We also thank King's College London for the use of the CREATE research computing facility (King's College London, 2022). Furthermore, the authors acknowledge the use of resources provided by the Isambard-AI National AI Research Resource (AIRR). Isambard-AI is operated by the University of Bristol and is funded by the UK Government's Department for Science, Innovation and Technology (DSIT) via UK Research and Innovation; and the Science and Technology Facilities Council [ST/AIRR/I-A-I/1023] (McIntosh-Smith et al., 2024). Access to these resources was granted under the AIRR Gateway project (Project ID: BYYG-VXGF-P).

# Impact Statement

This paper presents LSSAR, a novel attention mechanism designed to improve the numerical stability and length extrapolation capabilities of large language models. As a contribution to foundational models, our work has broad implications for the field of machine learning.

**Positive Impacts.** LSSAR's improved length extrapolation enables more efficient processing of long documents, potentially reducing computational costs and energy consumption compared to methods that require retraining on longer sequences. The enhanced numerical stability may also improve the reliability of LLM deployments in safety-critical applications. Furthermore, our symbolic regression experiments suggest that LSSAR can help models learn more physically grounded representations, which may be useful for scientific modelling applications.

Beyond these direct benefits, LSSAR implements a *coarse-to-fine causal filtering* mechanism (see Appendix A.2.1 for theoretical details): the Shift-ReLU stage performs coarse-grained noise elimination by categorically removing tokens below the population average, while the subsequent WTA dynamics perform fine-grained causal selection among surviving candidates. This hierarchical filtering has two promising implications. First, as discussed in Appendix B.7.1, it is particularly relevant for reasoning models that rely on Chain-of-Thought (CoT) processes. By progressively distilling relevant premises from noise, LSSAR may reduce the redundant self-verification loops observed in current reasoning systems, potentially compacting reasoning chains and improving token efficiency. Second, this capability opens avenues for AI safety research: (a) the coarse filtering stage may enhance robustness against adversarial attacks that exploit attention dispersion by eliminating obviously irrelevant distractors; and (b) the sparse, interpretable attention patterns resulting from fine filtering (see Appendix B.5) could facilitate AI alignment efforts by making the model's causal reasoning more transparent. We believe these directions warrant further investigation.

**Potential Concerns.** As with any advancement in LLM capabilities, LSSAR could be misused to generate harmful content more effectively over longer contexts. However, we believe this risk is not fundamentally different from those posed by existing LLM technologies, and the same mitigation strategies (content filtering, responsible deployment practices) remain applicable.

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

# A. Theoretical Analysis

## A.1. Detailed Algorithm of LSSAR

To help readers better understand the theoretical analysis in this appendix without needing to repeatedly refer back to the main text for equations, we provide the detailed algorithmic procedure of LSSAR in Algorithm 1. This self-contained reference explicitly presents all mathematical operations involved in each stage.

---

**Algorithm 1** Detailed Computation of LSSAR

---

**Require:** $\mathbf{Q}, \mathbf{K} \in \mathbb{R}^{L \times d}$: Query and Key matrices
**Require:** $\mathbf{M}' \in \{0, 1\}^{L \times L}$: Causal mask
**Require:** $p > 0$: Power parameter
**Ensure:** $\mathbf{A} \in \mathbb{R}^{L \times L}$: Sparse attention matrix

    **Stage 1: Normalisation (LSSA)**
1:   $\mathbf{Q}_i \leftarrow \mathbf{Q}_i / \|\mathbf{Q}_i\|_2$                    $\triangleright\ l_2$-normalise queries
2:   $\mathbf{K}_i \leftarrow \mathbf{K}_i / \|\mathbf{K}_i\|_2$                    $\triangleright\ l_2$-normalise keys
3:   $\mathbf{N}_{ij} \leftarrow i$                            $\triangleright\ \mathbf{N}$: token count matrix
4:   $\mathbf{S} \leftarrow (\log d \cdot \log \mathbf{N}) \otimes \mathbf{Q}\mathbf{K}^T$     $\triangleright\ \mathbf{S}$: length-scaled scores
5:   $\mathbf{A} \leftarrow \mathrm{Softplus}(\mathbf{S}) \otimes \mathbf{M}'$     $\triangleright$ Apply activation and mask
6:   $\mathbf{A}_i \leftarrow \mathbf{A}_i / \|\mathbf{A}_i\|_1$             $\triangleright$ First $l_1$-normalisation

    **Stage 2: Sharpening (Re-weighting)**
7:   $\mathbf{O}_{ij} \leftarrow \mathbf{1}_{[i>3]}$               $\triangleright\ \mathbf{O}$: offset (0 for first 3 rows)
8:   $\mathbf{A} \leftarrow \mathrm{ReLU}^p(\mathbf{A} \cdot \mathbf{N} - \mathbf{O})$     $\triangleright$ Shift-ReLU sharpening
9:   $\mathbf{A}_i \leftarrow \mathbf{A}_i / \|\mathbf{A}_i\|_1$             $\triangleright$ Second $l_1$-normalisation

10:   **return A**

---

## A.2. From Soft Inhibition to Sparse Coding

Beyond the mathematical optimisation dynamics, the role of the $l_1$-norm in attention mechanisms can be reinterpreted through the lens of computational neuroscience. In biological neural networks, energy efficiency is a fundamental constraint; the brain consumes a disproportionate amount of the body's metabolic energy, yet individual neurons operate under a regime of sparse coding (Olshausen & Field, 1996). Most neurons remain silent at any given moment, firing action potentials only when specific, salient stimuli are present. This principle suggests that an ideal attention mechanism should not merely distribute weights, but actively suppress irrelevant signals to conserve computational 'energy' and maximise the signal-to-noise ratio.

The role of Softmax attention can be rigorously understood through the lens of Divisive Normalisation, a canonical neural computation proposed by Heeger (1992) to describe the nonlinear response properties of neurons in the primary visual cortex (V1). The Divisive Normalisation model computes a neuron's response as $R_i = \frac{f(E_i)}{\sigma + \sum_j f(E_j)}$, where $E_i$ is the excitatory input and $f(\cdot)$ is a nonlinear gain function. Comparing this with the Softmax formulation in Eq. (2), we observe a striking mathematical correspondence: both express a ratio where the numerator is a nonlinearly transformed activation of a single neuron (or position), and the denominator is the aggregate of all competing neurons' activations. This reveals that Softmax Attention is essentially a special case of neural competition, where the excitatory input undergoes an exponential nonlinear gain transformation $f(x) = e^x$, followed by lateral inhibition mediated by the $l_1$-norm denominator. Biologically, this mirrors the function of inhibitory interneurons, which suppress the activity of surrounding neurons, making the most active neurons more salient. However, Softmax fundamentally creates a 'dense code': due to the nature of the exponential function, no weight is ever exactly zero. In neuroscientific terms, this corresponds to a system with a high spontaneous firing rate or background noise. While this ensures differentiability, it violates the principle of metabolic efficiency and fails to filter out off-manifold noise, leading to the susceptibility to distractors discussed in previous sections.

LSSAR advances this paradigm significantly, functioning as a unified computational model of cortical microcircuits that

integrates multiple biological mechanisms into a single differentiable operator:

**Stage 1: Gain Control via Divisive (Shunting) Inhibition.** The first stage directly mirrors the classic Divisive Normalisation model. It stabilises signal magnitude by normalising each neuron's response relative to the pooled activity of its neighbours, effectively implementing shunting inhibition. Crucially, the $l_1$-norm in this stage serves more than preliminary gain control; its primary function is to enforce *global competition* among all neurons. By coupling all inputs via the denominator, it ensures that every neuron, even those destined for suppression by subsequent thresholding via Shift-ReLU, contributes to the gradient calculation during backpropagation. This gradient coupling is the key mechanism that prevents the dead neuron problem: if a suppressed neuron needs to become active to lower the loss, the gradient flows back through the normalisation term to push its pre-activation value up, creating a robust learning dynamic without requiring slow homeostatic regulation.

**Stage 2: Noise Filtering and Decision Sharpening.** The second stage integrates two complementary biological mechanisms. First, the Shift-ReLU mechanism functionally mimics the rheobase or voltage threshold in biological neurons: a neuron does not fire unless its input current exceeds a dynamic baseline, represented here by the mean activity. This implements *subtractive inhibition* (Carandini & Heeger, 2012), where a uniform inhibitory signal (the local average) is subtracted from all inputs before a hard threshold is applied. By assigning exact zero probability to tokens falling below this threshold, LSSAR transitions from a dense, noisy representation to a true sparse code. This parallels the 'all-or-none' nature of biological action potentials, where sub-threshold stimuli result in silence. The Shift-ReLU mechanism ensures true metabolic efficiency via sparsity: if the input stimulus is weak or uniform, the mean-shift operation effectively suppresses all activity to zero, simulating the silence of a neuron in the absence of a driving stimulus.

Second, the power operation ($\mathrm{ReLU}^p$) followed by re-normalisation implements *recurrent lateral inhibition* in a feed-forward manner. In biological circuits, ambiguity among competing stimuli is resolved through iterative cycles of mutual inhibition until a winner emerges, the well-known Winner-Take-All (WTA) dynamics (Douglas et al., 1995). Our power parameter $p$ mathematically simulates the steady-state distribution of this recurrent process: a higher $p$ corresponds to more rounds of recurrent competition, sharpening the decision without the computational cost of iterative unrolling. The second $l_1$-norm, applied after the power operation, functions as the convergence mechanism for this recurrent competition, ensuring the output remains a valid probability distribution representing the WTA outcome.

The separation of the two $l_1$-norm stages is motivated primarily by engineering considerations rather than biological constraints. The first $l_1$-norm establishes gradient connectivity and global competition *before* the decision-making process begins, while the second $l_1$-norm implements the competitive dynamics that sharpen the final decision. This architectural choice allows LSSAR to maintain robust gradient flow (preventing dead neurons) while still achieving sharp, sparse attention distributions, which single-stage mechanisms cannot achieve. While this design incidentally mirrors certain aspects of cortical processing (e.g., the temporal separation between gain control and decision-making), the primary justification is optimisation stability rather than biological plausibility.

Beyond the engineering rationale, there is an interesting parallel to the semi-saturation constant $\sigma$ in the standard Divisive Normalisation model, which handles low-stimulus regimes. In LSSAR, the Shift-ReLU mechanism serves a functionally analogous role as a hard dynamic threshold: when input stimuli are weak or uniform, the mean-shift operation suppresses activity to zero, while the power parameter $p$ allows each attention head to tune its selectivity or sharpening based on task demands. However, this analogy should be understood as an observation rather than a design principle.

This transition from dense to sparse coding offers distinct advantages in high-dimensional sensory processing, akin to how the visual cortex uses sparse coding to disentangle explanatory factors from noise. LSSAR's hard sparsity acts as a denoising filter, preventing the aggregation of numerous weak, irrelevant signals, identified as 'manifold drift' that confuse standard Softmax models (see Appendix A.4 for details). From this perspective, LSSAR is not merely an engineering improvement but a step towards a more biologically plausible attention mechanism that unifies gain control, noise filtering, and competitive decision-making within a single differentiable framework.

### A.2.1. COARSE-TO-FINE CAUSAL FILTERING AND SYMBOLIC REGRESSION

The biological mechanisms described above can be unified into a coherent framework of *coarse-to-fine causal filtering*, which provides a plausible explanation for why LSSAR can improve symbolic-regression outcomes in our experiments (Sect. 3.4). In particular, the recovered inverse-square dependence suggests that this filtering process can help isolate the compact physical structure present in orbital trajectories.

**Coarse Filtering via Shift-ReLU.** The Shift-ReLU mechanism performs *coarse-grained noise elimination*. By subtracting

the mean activity and applying a hard threshold, this stage categorically removes tokens whose relevance falls below the population average. In the context of orbital trajectory sequences, this may correspond to down-weighting redundant or weakly informative time steps, local trajectory fluctuations, and numerical noise that do not strongly constrain the force relation. This coarse filtering reduces the search space for the subsequent refinement stage, transforming a dense, noisy attention distribution into a sparse set of candidate causal variables.

**Fine Filtering via WTA Dynamics.** The power operation implements *fine-grained causal selection* among the surviving candidates. As discussed earlier, this operation mathematically simulates $p$ iterations of recurrent lateral inhibition, progressively amplifying the strongest signals while suppressing weaker competitors. Physical relationships, such as the inverse-square dependence $F \propto m/r^2$, are characterised by sparse functional structure: the gravitational force depends on a small set of variables (mass and distance), not on the myriad of irrelevant contextual features present in raw trajectory data. The WTA dynamics may help filter candidate dependencies by amplifying compact, structured relations while suppressing spurious correlations that survived the coarse filtering stage.

This coarse-to-fine causal filtering mirrors how the brain's cortical circuits perform perceptual inference: an initial rapid feedforward sweep eliminates obviously irrelevant stimuli (analogous to Shift-ReLU), followed by iterative recurrent processing that resolves ambiguity among competing hypotheses (analogous to the power operation). Standard Softmax attention, with its dense, non-sparse distributions, lacks both filtering stages: it aggregates information from all tokens indiscriminately, injecting noise that obscures sparse underlying relationships.

## A.3. Comparison with Projection-Based Sparse Mechanisms

In the main text, we addressed the limitations of Softmax regarding attention smoothing and numerical instability. It is worth noting that a distinct family of probability mapping functions, most notably Sparsemax (Martins & Astudillo, 2016) and Entmax (Peters et al., 2019), has also been proposed to induce sparsity in attention distributions. While these methods share our motivation of mitigating noise and improving interpretability by assigning zero probability to irrelevant tokens, they rely on a fundamentally different mathematical foundation: using Euclidean projection onto the probability simplex. This creates specific challenges for large-scale language modelling compared to our proposed LSSAR mechanism.

The primary distinction lies in the computational complexity and hardware efficiency. Sparsemax and its generalised form, Entmax, require computing a threshold that involves sorting or iterative bisection of the input vector elements. This sorting operation typically has a complexity of $\mathcal{O}(L \log L)$ or $\mathcal{O}(L)$, and is computationally expensive on modern accelerators (GPUs/TPUs) compared to matrix multiplications and element-wise operations. More critically, the requirement for global sorting or thresholding across the sequence dimension hinders compatibility with I/O-aware optimisations such as FlashAttention. FlashAttention relies on tiling techniques that process blocks of the attention matrix in fast SRAM without materialising the full matrix. Since Sparsemax requires global knowledge of the row to determine the projection threshold, it cannot be easily fused into tiled kernels. In contrast, our LSSAR mechanism relies on element-wise operations (Softplus, ReLU, and power functions) and row-wise normalisation. These operations are local or reducible, suggesting that LSSAR may be amenable to future tiled or fused attention implementations. However, we have not yet demonstrated such an optimised kernel, and the additional operations still introduce non-trivial computational overhead in the current implementation.

Furthermore, the training dynamics of LSSAR differ significantly from projection-based methods. Both Sparsemax and direct ReLU-based attention mechanisms introduce exact zeros into the probability distribution, creating "dead zones" where the gradient is strictly zero. This 'dead neuron' problem fundamentally impedes gradient flow: once a token's score falls below the threshold, it loses all gradient feedback and becomes permanently isolated from the optimisation process.

LSSAR resolves this problem through its two-stage architecture. The key insight is that the $l_1$-normalisation in Stage 1 establishes gradient connectivity *before* the sparsity-inducing operations in Stage 2. By coupling all tokens through the shared denominator, even tokens that are subsequently suppressed to zero continue to receive gradient feedback, enabling their potential re-activation in future training steps. This design allows LSSAR to achieve sparse attention distributions while maintaining the robust gradient landscape essential for stable training. A detailed biological interpretation of this mechanism as divisive and subtractive inhibition is provided in Appendix A.2.

## A.4. Log-Space Formulation and Thermodynamic Interpretation

From an engineering perspective, we observe that the proposed re-weighting mechanism can be reformulated in the logarithmic space, revealing a direct equivalence to a temperature-scaled Softmax function. This insight allows for substantial acceleration by leveraging highly optimised standard kernels. Let us define the transformation $u_i = p \cdot \log(x_i)$, where cases of $x_i = 0$ are handled by assigning $u_i = -\infty$. Applying the standard Softmax function to the transformed scores **u** yields:

$$\text{Softmax}(\mathbf{u})_i = \frac{e^{u_i}}{\sum_j e^{u_j}} = \frac{e^{p \log x_i}}{\sum_j e^{p \log x_j}} \tag{A.1}$$

Utilising the logarithmic identity $e^{a \log b} = b^a$, the expression simplifies to the original power formulation:

$$\frac{(e^{\log x_i})^p}{\sum_j (e^{\log x_j})^p} = \frac{x_i^p}{\sum_j x_j^p} \tag{A.2}$$

This derivation proves that Stage 2 is mathematically identical to applying Softmax to the logarithm of the ReLU outputs scaled by $p$.

To rigorously justify the interpretation of $p$ as an *inverse temperature coefficient*, we can juxtapose our log-space formulation with the standard Boltzmann distribution (or Softmax with temperature $T$) used in statistical mechanics and machine learning:

$$\text{Standard Boltzmann:} \quad P(i) \propto \exp\left(\frac{z_i}{T}\right) \tag{A.3}$$

$$\text{LSSAR Log-Space:} \quad P(i) \propto \exp\left(p \cdot \log x_i\right) \tag{A.4}$$

By mapping the logarithmic scores $\log x_i$ to the energy states (logits) $z_i$, the structural alignment becomes evident:

$$p \equiv \frac{1}{T} \tag{A.5}$$

While standard attention mechanisms often operate at a fixed temperature (e.g., $T = \sqrt{d}$), the LSSAR framework employs $p$ to dynamically control the sharpness of the distribution. A large power parameter (e.g., $p = 15$) corresponds to a regime of extremely low temperature ($T \approx 0.06$). As $p \to \infty$ (or $T \to 0$), the system approaches its ground state, where the Softmax function asymptotically converges to the argmax operation. Consequently, the re-weighting mechanism can be understood as a process of active entropy minimisation, forcing the model to make decisive, low-uncertainty selections of relevant tokens. This thermodynamic view explains why increasing $p$ fundamentally strengthens the model's ability to maintain focus over long sequences, preventing the probability mass from diluting into a uniform distribution (high entropy) as the context window expands.

It is essential to highlight that the effectiveness of the re-weighting stage relies not only on the power parameter $p$ but also on the preceding ReLU activation. The shift operation ($\mathbf{A} \otimes \mathbf{N} - \mathbf{O}$) followed by ReLU functions as a hard noise filter or a dynamic hard mask. By truncating the tail of the distribution and assigning exact zero probability to tokens falling below the heuristic baseline, this step explicitly eliminates the majority of irrelevant context before the sharpening process begins.

This hard masking significantly alleviates the burden on the power operation. In a standard Softmax (infinite support), the temperature parameter must be extremely low (i.e. $p$ extremely high) to suppress the accumulated mass of thousands of irrelevant tokens to a negligible level. In LSSAR, since the ReLU has already removed the noise tokens, the power parameter $p$ serves a more refined purpose: it only needs to distinguish between the remaining potentially relevant candidates. This synergy allows LSSAR to achieve sharp, sparse attention distributions with a moderate finite $p$ (e.g., $p = 15$), avoiding the numerical instability and gradient vanishing problems often associated with the extreme temperature scaling required by standard attention mechanisms to achieve similar sparsity.

## A.5. Graph-Theoretic and Geometric Perspectives

This section provides a rigorous theoretical justification for the superiority of LSSAR over standard Softmax attention. Our analysis proceeds from two complementary perspectives. First, we employ the graph-theoretic framework proposed by (Wu et al., 2025) to demonstrate how LSSAR structurally eliminates the inherent position bias found in Transformers. Second, we interpret this structural correction through the lens of the manifold hypothesis to explain the model's enhanced length extrapolation capabilities.

### A.5.1. STRUCTURAL BIAS ELIMINATION VIA GRAPH TOPOLOGY

Recent theoretical advancements by (Wu et al., 2025) have modelled the flow of information in Transformers as a directed graph $\mathcal{G}$, where tokens represent nodes and attention scores represent weighted edges. Under the standard causal masking setting, the first token in a sequence acts as a unique 'centre node': the only node that is a predecessor to all other tokens. (Wu et al., 2025) proved that when the attention mechanism utilises the Softmax function, which assigns a strictly positive probability ($A_{ij} > 0$) to all allowed connections, the attention distribution inevitably collapses towards this centre node as the network depth increases. This phenomenon, often termed the 'attention sink', is not necessarily driven by semantic relevance but is a structural artefact of the full connectivity within the causal graph. Consequently, deep Transformers exhibit a systematic position bias, disproportionately attending to the initial tokens regardless of the input context.

LSSAR fundamentally resolves this structural pathology by altering the topology of the attention graph. By introducing the ReLU in the re-weighting stage, LSSAR enforces hard sparsity, allowing attention weights $A_{ij}$ to be exactly zero. This operation effectively severs the edges in the attention graph that fall below the adaptive mean-based threshold. Mathematically, this violates the strictly positive connectivity assumption required for the convergence results in (Wu et al., 2025). By dynamically disconnecting the edges between current tokens and irrelevant predecessors (including the initial token when it serves no semantic purpose), LSSAR prevents the probability mass from accumulating at the sequence start. Thus, the model is liberated from the structural position bias, allowing it to distribute attention solely based on semantic relevance.

### A.5.2. GEOMETRIC INTERPRETATION: PREVENTING MANIFOLD DRIFT

While the graph-theoretic view explains the elimination of bias, the manifold hypothesis (Bengio et al., 2013) offers a geometric explanation for the robustness of LSSAR, which parallels recent findings in architectural design such as Manifold-Constrained Hyper-Connections (mHC) (Xie et al., 2025). The mHC framework addresses the instability of residual connections in deep networks by projecting the weight matrices onto the Birkhoff polytope (the manifold of doubly stochastic matrices). This geometric constraint strictly preserves the norm and ensures that the residual mapping remains close to an identity function, effectively preventing signal explosion and drift during propagation.

Analogously, LSSAR enforces a topological manifold constraint on the attention graph to prevent what we term 'manifold drift'. In standard Softmax attention, the mechanism aggregates information from the entire context window. As the sequence length extends, the cumulative contribution of numerous irrelevant tokens grows significantly. This off-manifold noise pulls the aggregated hidden states away from the intrinsic semantic manifold. This corresponds to the unconstrained instability observed in hyper-connections, where unrestricted mixing leads to representation collapse.

By introducing the Shift-ReLU mechanism, LSSAR acts as a manifold-aware projection operator. The hard sparsity constraint functions similarly to the boundary conditions of the Birkhoff polytope in mHC: it strictly filters out off-manifold noise vectors that fall below the adaptive threshold. By constructing a sparse, dynamic $k$-Nearest Neighbours ($k$-NN) graph rather than a fully connected dense graph, LSSAR ensures that the representation update is derived exclusively from a local neighbourhood of semantically relevant tokens. This preservation of local topology ensures that the hidden states remain firmly anchored to the semantic manifold, irrespective of the sequence length. Consequently, LSSAR achieves robust length extrapolation by fundamentally preventing the representations from drifting into undefined regions of the latent space, just as mHC ensures stability by constraining residuals to a well-defined geometric region.

### A.6. Asymptotic Gradient Behaviour

In this section, we provide a rigorous analysis of the optimisation dynamics by examining the asymptotic behaviour of the gradients in the saturation regime. We demonstrate that the standard Softmax function suffers from exponential gradient decay, which fundamentally conflicts with the sharpening objective of the re-weighting mechanism. In contrast, we show that the LSSAR formulation exhibits polynomial gradient decay, maintaining a robust learning signal even under high sharpening factors.

To analyse the gradient flow, we trace the derivative from the re-weighted output back to the input logit $x$. The LSSAR pipeline introduces an intermediate filtering stage (Shift-ReLU) between the generator and the power operation. Let $a(x)$ be the normalised generator output, $r(a) = \text{ReLU}(a - \mu)$ be the filtered score, and $y = r^p$ be the final sharpened output. The total gradient is given by the chain rule:

$$\frac{\partial y}{\partial x} = \frac{\partial y}{\partial r} \cdot \frac{\partial r}{\partial a} \cdot \frac{\partial a}{\partial x} \tag{A.6}$$

We focus our analysis on the 'winner' token regime (where $x_i \to \infty$). In this active regime, the token survives the filtering stage ($a > \mu$), meaning the ReLU operates in its linear region. Consequently, the filter gradient is $\frac{\partial r}{\partial a} \approx 1$. The dynamics are thus dominated by the interaction between the amplifier ($p \cdot r^{p-1}$) and the source gradient.

For the standard Softmax mechanism, the generator is the Softmax function. The derivative of the Softmax function with respect to its input is given by $\frac{\partial a_i}{\partial x_i} = a_i(1 - a_i)$. To rigorously substantiate the exponential decay of this source gradient, we expand the derivation of the term $(1 - a_i)$. By isolating the contribution of the target token $x_i$ from the aggregate score of all other tokens, denoted as $C = \sum_{j \neq i} e^{x_j}$, the expression for the complement probability becomes:

$$1 - a_i = 1 - \frac{e^{x_i}}{e^{x_i} + C} = \frac{(e^{x_i} + C) - e^{x_i}}{e^{x_i} + C} = \frac{C}{e^{x_i} + C} \tag{A.7}$$

In the regime where the model assigns high confidence to the winner token (i.e., as $x_i \to \infty$), the exponential term $e^{x_i}$ grows significantly larger than the constant interference term $C$. Consequently, the denominator is dominated by $e^{x_i}$, allowing us to approximate the expression as $\frac{C}{e^{x_i}} = C \cdot e^{-x_i}$. Since $\lim_{x_i \to \infty} a_i = 1$, the magnitude of the full derivative $a_i(1 - a_i)$ is entirely dominated by this decay term:

$$\frac{\partial a_i}{\partial x_i} \approx 1 \cdot (C \cdot e^{-x_i}) \propto e^{-x_i} \tag{A.8}$$

This exponential decay presents a fatal obstacle for the re-weighting mechanism. The total gradient becomes the product of a polynomial term (from the power operation) and an exponential decay term. Since exponential decay dominates polynomial growth for any finite $p$, the total gradient vanishes rapidly as the model becomes confident. This phenomenon, which we term *exponential saturation*, effectively locks the weights and prevents further optimisation.

In stark contrast, LSSAR employs Softplus followed by $l_1$-normalisation. The derivative of the Softplus function is the Sigmoid function, $\sigma(x) = (1 + e^{-x})^{-1}$. As $x_i \to \infty$, $\sigma(x_i)$ asymptotically approaches 1. This upper bound signifies that the generator itself does not vanish or saturate in the high-confidence regime, unlike the exponential tail of Softmax. Consequently, Softplus($x_i$) approaches linearity ($\approx x_i$), and the normalised score approximates a harmonic function $a_i \approx x_i/(x_i + K)$, where $K$ represents the contribution of other tokens. The derivative of this function follows a polynomial decay:

$$\frac{\partial a_i}{\partial x_i} \approx \frac{1 \cdot (x_i + K) - x_i \cdot 1}{(x_i + K)^2} = \frac{K}{(x_i + K)^2} \propto x_i^{-2} \tag{A.9}$$

Crucially, because the intermediate Shift-ReLU operation behaves linearly for active tokens, it preserves this polynomial characteristic. The total gradient is thus the product of a polynomial growth term (from the power operation) and a polynomial decay term (from the source). Unlike the Softmax case, these terms can balance each other or even result in net gradient amplification. This structural property ensures that the optimisation pathway remains open even in the high-confidence regime, explaining why LSSAR can be stably trained with high sharpening factors where standard Softmax architectures fail.

## B. Further Experiments

All models were trained using a sequence length of 1024 on the FineWeb-10B dataset. We used GPT-2-124M (12 layers, 12 heads, and hidden size 768) with RoPE, Adam with $\beta_1 = 0.9$ and $\beta_2 = 0.95$, a learning rate of $6 \times 10^{-4}$, 700 warmup steps, cosine decay, weight decay 0.1, and gradient clipping at 1.0. Training used 8 NVIDIA A100 80GB GPUs, a micro-batch size of 4 per GPU, gradient accumulation of 16, and PyTorch Automatic Mixed Precision (AMP) with bfloat16 input precision. The resulting batch size was 524,288 tokens per step, and the full run lasted 18,865 iterations, corresponding to approximately 9.89B training tokens.

Unless otherwise specified, all experiments outside the dedicated physical-law and scaling analyses use this FineWeb-10B training setup with the GPT-2-124M architecture. This includes the ablation studies, Softmax-free attention comparisons, long-context extrapolation experiments, and attention-score visualisations.

All baselines used the same training and evaluation setup; only the attention mechanism was changed.

*Table A.4.* Validation loss values for training the GPT-2 model from scratch with the standard and modified attention scores.

|       | Softmax | Inverse | Re-centred | Re-centred & $l_1$-norm |
|-------|---------|---------|------------|-------------------------|
| Loss  | 3.1911  | 3.1915  | 3.2008     | 3.1954                  |

*Table A.5.* Validation loss for GPT-2 model with and without the Softmax components: $e^x$ and $l_1$-norm.

| Scenario    | I      | II     | III    | IV     |
|-------------|--------|--------|--------|--------|
| $e^x$       |        | ✓      |        | ✓      |
| $l_1$-norm  |        |        | ✓      | ✓      |
| Loss        | 3.4138 | 3.247  | 3.3297 | 3.1911 |

## B.1. Ablation Study

### B.1.1. MODIFIED ATTENTION SCORES

To investigate the necessity of non-negative attention scores, we conducted experiments with modified Softmax outputs. As shown in Tab. A.4, inverting the Softmax outputs by multiplying them by negative one resulted in a negligible change in validation loss, suggesting that non-negativity alone is not the determining factor. We also re-centred the attention scores by subtracting the row-wise mean, creating a mix of positive and negative values[1]. This led to a minor increase in loss. However, when we subsequently scaled these re-centred scores by the $l_1$-norm, performance was almost fully restored. Together with the decomposition study in Tab. A.5, these results suggest that Softmax's effectiveness comes from the combination of a nonlinear transformation and subsequent normalisation, with the $l_1$-norm playing a central role in preserving performance.

### B.1.2. SOFTMAX DECOMPOSITION

As illustrated in Eq. (2), the Softmax operation can be decomposed into a non-linear transformation ($e^x$), followed by the $l_1$-norm. In this section, we examine the importance of each component for LLMs by training the GPT-2 model from scratch with and without each component.

The results presented in Tab. A.5 provide quantitative evidence for our theoretical analysis detailed in Sect. 2.2.1. First, the comparison between Scenario III and Scenario IV highlights the critical role of the non-linear transformation. Scenario III, which applies the $l_1$-norm directly to raw attention scores (effectively a form of linear attention), results in a higher validation loss than Scenario IV. This indicates that without the expansive mapping provided by the exponential function, the model struggles to distinguish between relevant and irrelevant tokens, leading to an overly flat attention distribution.

Most critically, the comparison between Scenario II and Scenario IV empirically substantiates the vital role of the $l_1$-norm as a mechanism for *lateral inhibition* and *gradient coupling*. While better than the linear baseline, Scenario II still lags behind the full Softmax. This performance gap demonstrates that non-linearity alone is insufficient. Without the $l_1$-norm, the attention mechanism lacks the 'zero-sum' constraint required to induce competition among tokens. Consequently, the superior convergence observed in Scenario IV confirms that the gradient coupling provided by the $l_1$-norm is indispensable for stabilising the optimisation process and preventing the isolation of token representations. These findings justify our design choice in LSSA to retain both a strong non-linear mapping (Softplus) and the $l_1$-norm to preserve these essential dynamical properties.

### B.1.3. COMPARISON WITH DIFFERENT ACTIVATION FUNCTIONS

We modify the standard attention mechanism by introducing several novel attention variants for comparative analysis. Specifically, these variants are created by substituting $\phi$ in Eq. (3) with several alternative activation functions: ReLU (Rumelhart et al., 1986), ReLU$^2$ (So et al., 2021), ReLU6 (Howard et al., 2017), GeLU (Hendrycks, 2016), Sigmoid (Rumelhart et al., 1986), Softplus (Zheng et al., 2015), and Mish (Misra, 2019).

The results presented in Tab. A.6 indicate that all activation functions, except for Softplus, result in poorer validation loss values compared to the standard attention mechanism represented by $e^x$. This performance disparity can be rigorously

---

[1]To prevent training instability due to the absence of non-zero values in the initial rows, the first three rows were left unmodified.

*Table A.6.* Validation loss for various activation functions employed in the attention mechanism. Note that $\phi = e^x$ corresponds to the conventional Softmax attention.

|        | ReLU   | ReLU$^2$ | ReLU6  | GeLU   | Sigmoid | Softplus   | Mish   | $e^x$  |
|--------|--------|----------|--------|--------|---------|------------|--------|--------|
| Loss   | 3.2006 | 3.2494   | 3.2039 | 3.2051 | 3.2000  | **3.1901** | 3.2001 | 3.1911 |

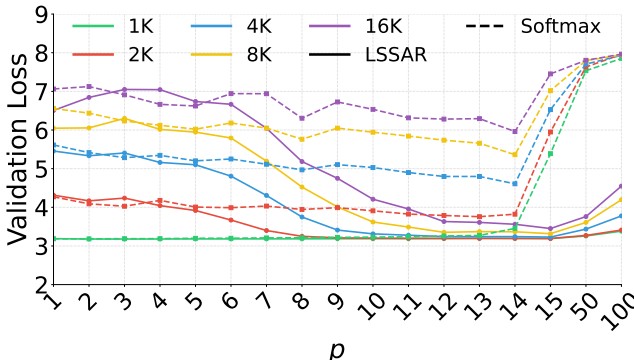

*Figure A.3.* Comparison of LSSA and Softmax attention with varying values of $p$ for the re-weighting mechanism across different sequence lengths.

explained through our theoretical framework regarding gradient coupling and mapping intensity.

First, ReLU-based functions (ReLU, ReLU$^2$, ReLU6) violate the principle of **non-zero participation**. By mapping negative inputs strictly to zero, these functions effectively remove a large portion of tokens from the $l_1$-norm denominator. As discussed in Sect. 2.2.1, this severs the gradient pathway for these suppressed tokens, creating 'dead neuron' that cannot be optimised via lateral inhibition. Consequently, the model loses the ability to recover potentially relevant information from the negative regime, leading to suboptimal convergence.

Second, while Sigmoid is strictly positive, it is bounded and saturates at both ends of the input range. When inputs enter these saturation regions, the gradient becomes small, making the attention mechanism less sensitive to variations in the latent representation. In contrast, Softplus and $e^x$ remain unbounded on the positive side, which helps preserve gradient flow before normalisation.

In contrast, both Softplus and $e^x$ satisfy the two critical conditions for effective attention: they are strictly positive (ensuring global gradient coupling via the $l_1$-norm) and unbounded (preventing saturation). Softplus, in particular, achieves the lowest validation loss. This slight advantage over $e^x$ may be attributed to its asymptotic linearity in the positive domain, which provides a more stable gradient flow compared to the exponential growth of $e^x$, thereby balancing the need for feature separation with numerical stability.

### B.1.4. VARYING $p$ FOR RE-WEIGHTING MECHANISM

To gain deeper insights into the proposed re-weighting mechanism, we evaluated it using LSSA and Softmax attention across a range of $p$. As illustrated in Fig. A.3, LSSA and Softmax attention exhibit distinct behaviours to changes in the re-weighting parameter $p$. While Softmax attention performs comparably to LSSA at lower $p$ values, it suffers from severe gradient explosion issues as $p$ increases to $p = 15$ and beyond. This instability manifests as a sharp rise in validation loss across all sequence lengths, rendering the model practically unusable at these higher $p$ values.

In contrast, LSSA remains stable over the tested range of $p$ values. Increasing $p$ initially sharpens the re-weighted attention distribution and improves validation loss, with the best performance in this sweep observed around $p = 15$. However, we do not view this value as a quantity derived from first principles. Very large $p$ can over-concentrate probability mass on a small number of entries, so $p$ should be treated as a tunable hyperparameter selected by validation performance for the target model and context length.

In practice, the preferred value of $p$ depends on both the training sequence length and the model architecture. In our experiments, longer training lengths tended to require larger $p$ values. As provisional starting points for tuning, we suggest $p \approx 13$ for length 1024, $p \approx 18$ for length 2048, and $p \approx 25$ for length 4096. These values are empirical heuristics rather

*Table A.7.* Validation loss comparison between the base LSSAR model and the corresponding LSSAR+PI model after continued finetuning at each target context length. Lower is better.

| Target length | Base LSSAR ($p = 15$) | LSSAR ($p = 15$) + PI |
|---|---|---|
| 2K | 3.1930 | **3.1636** |
| 4K | 3.2291 | **3.1560** |
| 8K | 3.3171 | **3.1638** |

*Table A.8.* Passkey retrieval accuracy comparison between the base LSSAR model and the corresponding LSSAR+PI model after continued finetuning at each target context length. Accuracy is averaged over 100 trials. Higher is better.

| Target length | Base LSSAR ($p = 15$) | LSSAR ($p = 15$) + PI |
|---|---|---|
| 2K | 34% | **83%** |
| 4K | 15% | **48%** |
| 8K | 3% | **21%** |

than a universal closed-form rule. Additional adaptive-$p_i$ results are provided in Appendix B.3.

Crucially, while Softmax suffers from exponential gradient decay (gradients $\propto e^{-x}$) that causes optimisation to stall at high $p$, LSSAR maintains polynomial gradient decay ($\propto x^{-2}$) due to Softplus's bounded derivative (Sigmoid $\leq 1$). This ensures stable gradient flow even with aggressive sharpening (see Appendix A.6). Although validation loss increases gradually beyond $p = 15$ due to over-concentration on maximum values, LSSA handles large $p$ without numerical instability.

### B.2. Combined with Position Interpolation

To evaluate whether the proposed attention mechanism remains effective when combined with a standard positional long-context technique, we conducted an additional series of experiments using LSSAR together with Position Interpolation (PI). Starting from the 1024-token LSSAR checkpoint ($p = 15$), we performed continued finetuning with PI enabled at target context lengths of 2K, 4K, and 8K on the same FineWeb-10B dataset.

The continued finetuning setup was kept simple in order to isolate the effect of combining PI with the proposed attention mechanism. For each target length, we trained for 2000 additional steps using a learning rate of $6 \times 10^{-5}$, a 100-step warmup schedule, no further decay, and a total batch size of 524,288 tokens.

**Validation Loss.** The validation-loss results are reported in Tab. A.7. At all three target lengths, the LSSAR+PI model achieves lower validation loss than the corresponding base LSSAR checkpoint. The improvement becomes more pronounced as the target context length increases, indicating that PI complements the proposed attention mechanism in the continued long-context finetuning regime.

**Passkey Retrieval.** The Passkey Retrieval results are reported in Tab. A.8. Consistent with the validation-loss comparison, combining PI with LSSAR yields substantial gains at every target length. The largest relative improvements are observed on the passkey task, where the 2K, 4K, and 8K accuracies increase from 34% to 83%, from 15% to 48%, and from 3% to 21%, respectively. Although the absolute accuracy at 8K remains modest, this is likely attributable to the limited finetuning budget of 2000 steps from a 1024-token checkpoint rather than to an incompatibility between the two methods.

Taken together, these results provide direct empirical evidence that LSSAR remains effective when combined with PI, and that the gains from the proposed attention mechanism are preserved under a more practical long-context finetuning setup.

### B.3. Adaptive Position-Dependent $p_i$ Analysis

To further investigate whether the sharpening parameter should vary with sequence position, we evaluated a position-dependent adaptive rule for query row $i$:

$$p_i = \begin{cases} 15, & i \leq L_{\text{train}}, \\ 15 \cdot \left(1 + \alpha \left(\sqrt{i/L_{\text{train}}} - 1\right)\right), & i > L_{\text{train}}, \end{cases} \quad \text{(A.10)}$$

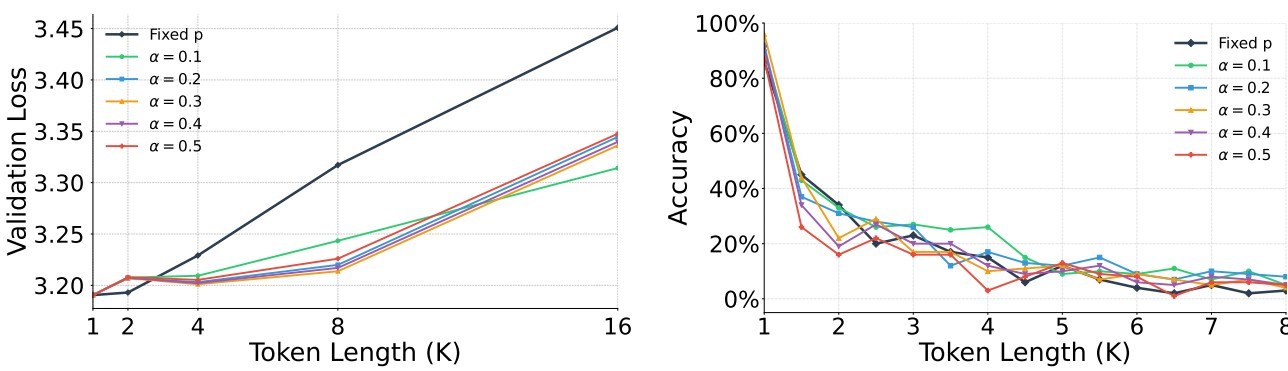

*Figure A.4.* Adaptive position-dependent $p_i$ analysis. Left: validation-loss extrapolation under different adaptive schedules. Right: passkey retrieval accuracy under the same schedules.

*Table A.9.* Computational overhead comparison for different attention mechanisms.

| Attention Mechanism | Training | | Evaluation | |
|---|---|---|---|---|
| | Time (ms) | Memory (MB) | Time (ms) | Memory (MB) |
| Standard Attention | 120.48 | 9609.45 | 49.26 | 2324.71 |
| LSSA | 169.92 | 12071.23 | 58.56 | 2325.58 |
| LSSAR ($p = 15$) | 250.32 | 16685.57 | 81.15 | 2325.58 |

where $L_{\text{train}} = 1024$ and $\alpha$ controls the extrapolation strength.

We evaluated $\alpha \in \{0.1, 0.2, 0.3, 0.4, 0.5\}$ and compared the resulting models with the fixed-$p = 15$ baseline. The results are shown in Fig. A.4. At extreme context lengths, the adaptive schedule improves validation loss. For example, at $16\times$ extrapolation, $\alpha = 0.1$ reduces the validation loss from 3.45 to 3.31. This indicates that allowing $p_i$ to increase gradually beyond the training horizon can partially improve perplexity-style extrapolation.

However, the corresponding Passkey Retrieval results degrade in a manner similar to the fixed-$p = 15$ setting, indicating that adaptive sharpening alone does not eliminate the dispersion issue. Taken together, these results suggest that a length-dependent $p_i$ schedule can improve validation-loss extrapolation, but does not by itself resolve the retrieval trade-off. We therefore present this adaptive analysis as a practical extension of the main hyperparameter study rather than as a complete solution.

### B.4. Computational Analysis

To evaluate the computational costs of our proposed methods, we conducted a benchmark analysis. The results, detailed in Table A.9, compare the performance of GPT-2-124M models with LSSA and LSSAR against the standard Softmax attention baseline. For a fair comparison, all methods were implemented using standard PyTorch functions without leveraging specialised fused CUDA kernels (e.g., FlashAttention). Experiments were conducted on a single NVIDIA A100 GPU using bfloat16 precision with a batch size of 4 and a sequence length of 1024.

The results in Table A.9 show that the current PyTorch implementation of LSSAR incurs substantial overhead relative to the standard Softmax baseline. In our benchmark, LSSAR ($p = 15$) increases evaluation latency from 49.26 ms to 81.15 ms and training memory from 9609.45 MB to 16685.57 MB. Its evaluation memory footprint remains nearly identical to the Softmax baseline, suggesting that the largest memory overhead arises in the backward pass, where the PyTorch Autograd engine must cache additional intermediate tensors for operations such as $\text{ReLU}^p$.

To examine whether an optimised implementation may be feasible, we analyse LSSAR's theoretical complexity. From a computational standpoint, the complexity of standard attention is dominated by two matrix multiplications, resulting in $O(L^2 d)$ floating-point operations (FLOPs). The additional operations in LSSAR (norms, element-wise functions) are of a lower order ($O(Ld)$ or $O(L^2)$), meaning that the asymptotic computational complexity of LSSAR remains identical to standard attention.

The more critical aspect for future optimisation is memory (I/O) complexity. The primary bottleneck in naive attention is the memory bandwidth required to read and write the large $L \times L$ attention matrix to and from High-Bandwidth Memory (HBM). I/O-aware algorithms like the FlashAttention family (Dao et al., 2022; Dao, 2023; Shah et al., 2024) solve this by computing the output in tiles without ever materialising the full matrix in HBM, reducing memory access complexity from $O(L^2)$ to the optimal $O(Ld)$. Crucially, all additional operations in LSSAR are local (element-wise or row-wise). This locality means they can be applied to a sub-block (tile) of the attention matrix within fast on-chip SRAM. Consequently, these operations may be amenable to fusion into the main loop of a tiled attention algorithm, although we have not yet demonstrated such a fused kernel in practice.

In summary, the locality of the additional operations suggests that LSSAR may be compatible with future tiled or fused implementations, but we have not yet demonstrated an optimised kernel or efficiency comparable to standard high-performance attention. The present discussion should therefore be viewed as preliminary feasibility analysis rather than a confirmed engineering result.

### B.5. Visualisation of Attention Scores

This section provides a visual comparison of attention maps generated by standard Softmax attention and our proposed LSSAR mechanism. Following the default configuration in Appendix B, the attention maps are extracted from the final layer of GPT-2-124M models trained on the FineWeb-10B dataset with a sequence length of 1024 tokens. The final layer is chosen for this analysis as its attention patterns are most indicative of the model's high-level understanding and directly influence the final output. Comparing these maps offers a clear view of how each mechanism synthesises information across the entire sequence.

As shown in Fig. A.1, the attention maps for all 12 heads produced by standard Softmax attention (left panel) exhibit the attention sink phenomenon. This is where the first token disproportionately attracts attention from other tokens, irrespective of its semantic importance. This issue is particularly pronounced in longer sequences, where the first token can dominate the attention distribution, leading to suboptimal model performance. In contrast, the attention maps generated by LSSAR (right panel) display a more balanced distribution of attention across tokens. Notably, in the 4x3 grid of LSSAR heads, those in the second column do not exhibit the attention sink phenomenon. This demonstrates that the proposed method effectively mitigates this problem, enabling the model to focus on more relevant tokens throughout the sequence. This visual evidence corroborates our quantitative findings, demonstrating that LSSAR not only enhances length extrapolation capabilities but also improves the overall quality of the attention distributions in transformer models.

### B.6. Experiments for Scaling with Filtered Data

To evaluate the robustness of the proposed LSSAR mechanism across different model scales and data distributions, we conducted a new series of experiments using the FineWeb-Edu dataset (Penedo et al., 2024). We specifically selected FineWeb-Edu over the standard FineWeb dataset used in the main text because it is rigorously filtered using Llama-3-70B, retaining only content with high educational value and logical coherence. This selection serves a dual purpose. On the one hand, it eliminates the potential influence of noisy training data on downstream task performance, ensuring that the evaluation reflects the intrinsic capabilities of the attention mechanism. On the other hand, training on such a clean, high-quality dataset tends to drive models to focus sharply on semantic and syntactic dependencies (Gunasekar et al., 2023). This characteristic paradoxically increases the difficulty of the Passkey Retrieval task, as the model must attend to a "passkey" token that acts as semantic noise within a highly coherent context. In this rigorous setting, the ability of an attention mechanism to distinguish and retrieve the passkey becomes a definitive test of its precision and extrapolation capabilities.

We trained three model configurations from scratch to investigate scaling behaviours. The first is a 6-layer GPT-2-45M model with a configuration suggested by the Pythia suite (Biderman et al., 2023) for analysing scaling behaviours. The rest are the standard GPT-2 architectures with 124M and 355M parameters respectively. All models were modified to incorporate RoPE to ensure a fair comparison with the state-of-the-art extrapolation baseline, and were trained on the FineWeb-Edu 100B dataset with a sequence length of 1024 tokens.

The GPT-2-124M experiments follow the default training configuration in Appendix B, except that the training corpus is replaced by FineWeb-Edu 100B. The GPT-2-45M and GPT-2-355M runs use the same sequence length, total batch size of 524,288 tokens, RoPE, weight decay, AMP/bfloat16 precision, and evaluation protocol. They differ from the GPT-2-124M setting only in the model scale and the corresponding optimisation schedule: GPT-2-45M uses a 6-layer configuration, a per-GPU batch size of 64, a learning rate of $1 \times 10^{-3}$, and 9,000 training steps; GPT-2-355M uses a 24-layer configuration,

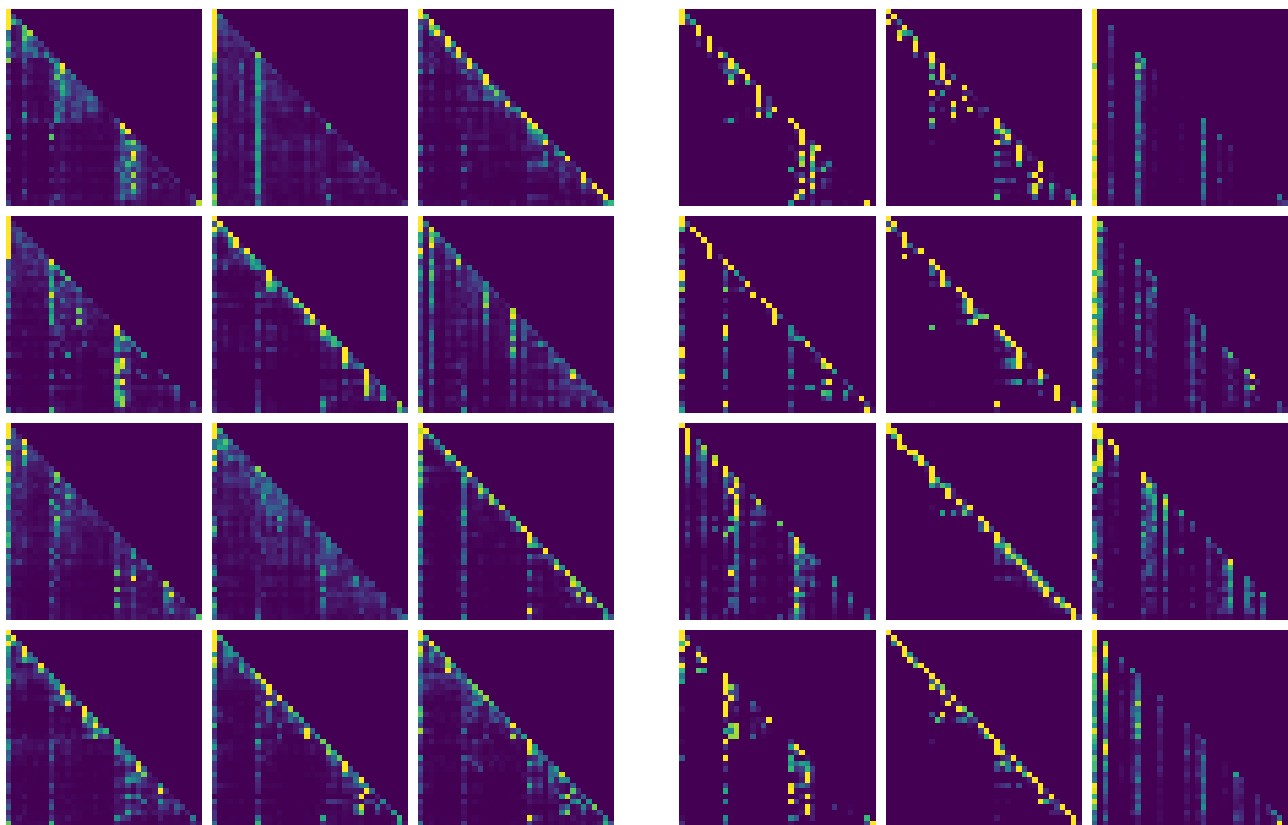

*Figure A.1.* Comparison of attention maps from the last layer of GPT-2-124M, showing standard Softmax attention (left) versus LSSAR with $p = 15$ (right). Each panel displays the 12 attention heads in a 4x3 grid. For visualisation purposes, attention scores are clamped to the range $[0, 0.5]$. The input text is: *Working from home can be great most days. I enjoy the flexibility and not having to commute in traffic. But sometimes I miss the office interactions with my coworkers and team meetings.*

a per-GPU batch size of 32, a learning rate of $3 \times 10^{-4}$, and 60,000 training steps.

**Validation Loss Extrapolation.** We first evaluated the language modelling performance on sequence lengths extending far beyond the training context. As illustrated in Figure A.2, the standard Softmax attention exhibits a significant degradation in validation loss as the sequence length increases, failing to extrapolate effectively even when trained on high-quality data. In contrast, LSSAR maintains a stable and nearly constant validation loss across all tested lengths for both the 45M and 355M models, demonstrating that its entropy invariance property holds true regardless of model scale or data quality.

**Passkey Retrieval Robustness.** The results for the Passkey Retrieval task are presented in Figure A.3. Consistent with our hypothesis regarding high-quality training data, the standard Softmax baseline fails completely, yielding 0% accuracy across all tested sequence lengths, including the training window itself. The model's strong bias towards coherent semantic

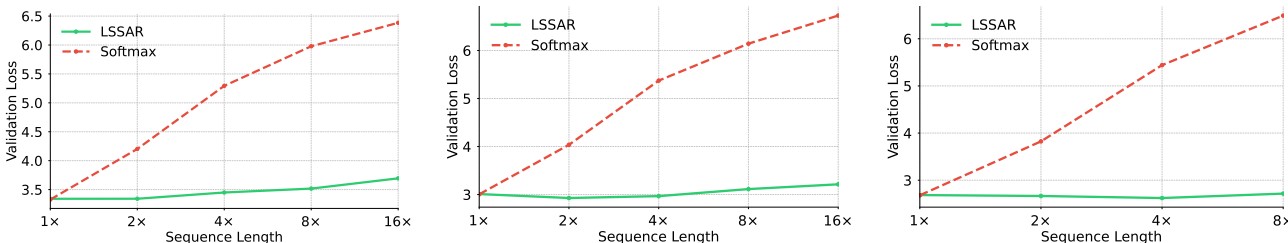

*Figure A.2.* Comparison of Softmax attention and LSSAR($p = 15$) with validation loss extrapolation for GPT-2-45M (left), GPT-2-124M (middle) and GPT-2-355M (right).

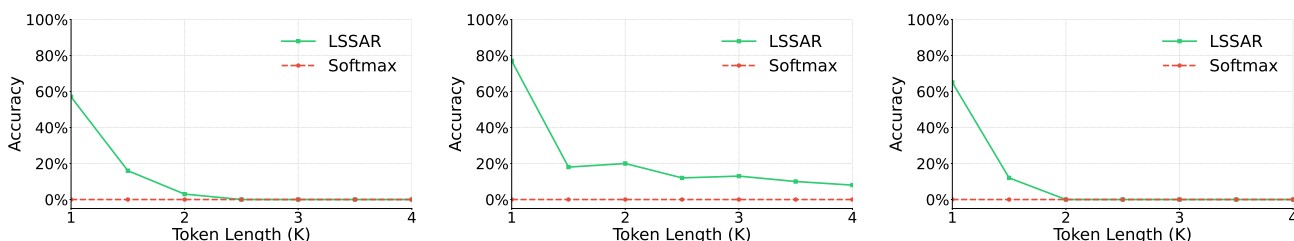

*Figure A.3.* Comparison of Softmax attention and LSSAR($p = 15$) with passkey retrieval accuracy for GPT-2-45M (left), GPT-2-124M (middle) and GPT-2-355M (right). Accuracy was averaged over 100 trials with the passkey placed at random positions within the sequence.

structures prevents it from attending to the random passkey, treating it effectively as noise. In stark contrast, LSSAR successfully overcomes this limitation, achieving substantial retrieval accuracy within the training length (57% for GPT-2-45M, 77% for GPT-2-124M and 65% for GPT-2-355M) and maintaining functional capabilities into the extrapolation regime. This result confirms that the proposed re-weighting mechanism effectively sharpens the attention distribution, enabling the model to capture critical high-entropy information even when trained on "textbook-quality" data that discourages such behaviour.

**Downstream Task Performance.** We evaluated zero-shot performance on standard downstream benchmarks. The results are detailed in Table A.1. LSSAR demonstrates superior performance compared to the Softmax baseline across the majority of tasks for all model scales (45M, 124M, and 355M). While Softmax shows marginal advantages in isolated cases (e.g., ARC-C and MMLU for the 124M model), LSSAR achieves significant gains in tasks requiring long-range dependency and knowledge retrieval. Notably, on the SummScreen benchmark, which requires processing long contexts for summarisation, LSSAR achieves substantial improvements over Softmax (approximately $2.7\times$ on 124M and $3.9\times$ on 355M). These gains confirm that the architectural improvements of LSSAR translate into tangible benefits for complex reasoning and summarisation tasks, without sacrificing generic capabilities.

*Table A.1.* Zero-shot performance on downstream tasks for models trained on FineWeb-Edu. Best scores are bolded.

| Model | Attention | ARC-E | ARC-C | HellaSwag | PIQA | MMLU | SciQ | SummScreen |
|---|---|---|---|---|---|---|---|---|
| GPT-2-45M | Softmax | 45.58 | 16.72 | 27.34 | 58.98 | 22.94 | 70.40 | 0.8100 |
| | LSSAR($p = 15$) | **46.04** | **18.34** | **27.49** | **59.79** | **22.97** | **75.00** | **2.1932** |
| GPT-2-124M | Softmax | 39.35 | **19.88** | 26.85 | 57.29 | **24.06** | 55.20 | 2.3313 |
| | LSSAR($p = 15$) | **44.49** | 19.11 | **29.46** | **65.89** | 22.96 | **70.60** | **6.2825** |
| GPT-2-355M | Softmax | 59.30 | 23.46 | 30.63 | 66.49 | 22.98 | 78.90 | 2.4506 |
| | LSSAR($p = 15$) | **62.50** | **26.96** | **34.81** | **68.34** | **23.94** | **84.00** | **9.5083** |

**Visualisation of Attention Scores.** Finally, we visualise the attention scores of all models using the same setting as in Appendix B.5. The results are presented in Fig. A.4, Fig. A.5 and Fig. A.6. The attention maps produced by standard Softmax attention exhibit the attention sink phenomenon. In contrast, the attention maps generated by LSSAR display a more balanced distribution of attention across tokens. Notably, in the 4x4 grid of LSSAR heads with GPT-2-355M, those in the first and third columns do not exhibit the attention sink phenomenon. This demonstrates that the proposed method effectively mitigates this problem, enabling the model to focus on more relevant tokens throughout the sequence. This visual evidence corroborates our quantitative findings, demonstrating that LSSAR not only enhances length extrapolation capabilities but also improves the overall quality of the attention distributions in transformer models.

### B.7. Additional Evaluation on LAMBADA

To further investigate the robustness of LSSAR against distribution shifts and its capability to maintain long-range coherence, we evaluated our models on the LAMBADA dataset (OpenAI split) (Paperno et al., 2016). This task requires the model to predict the last word of a passage, necessitating a deep understanding of the broader context rather than relying solely on

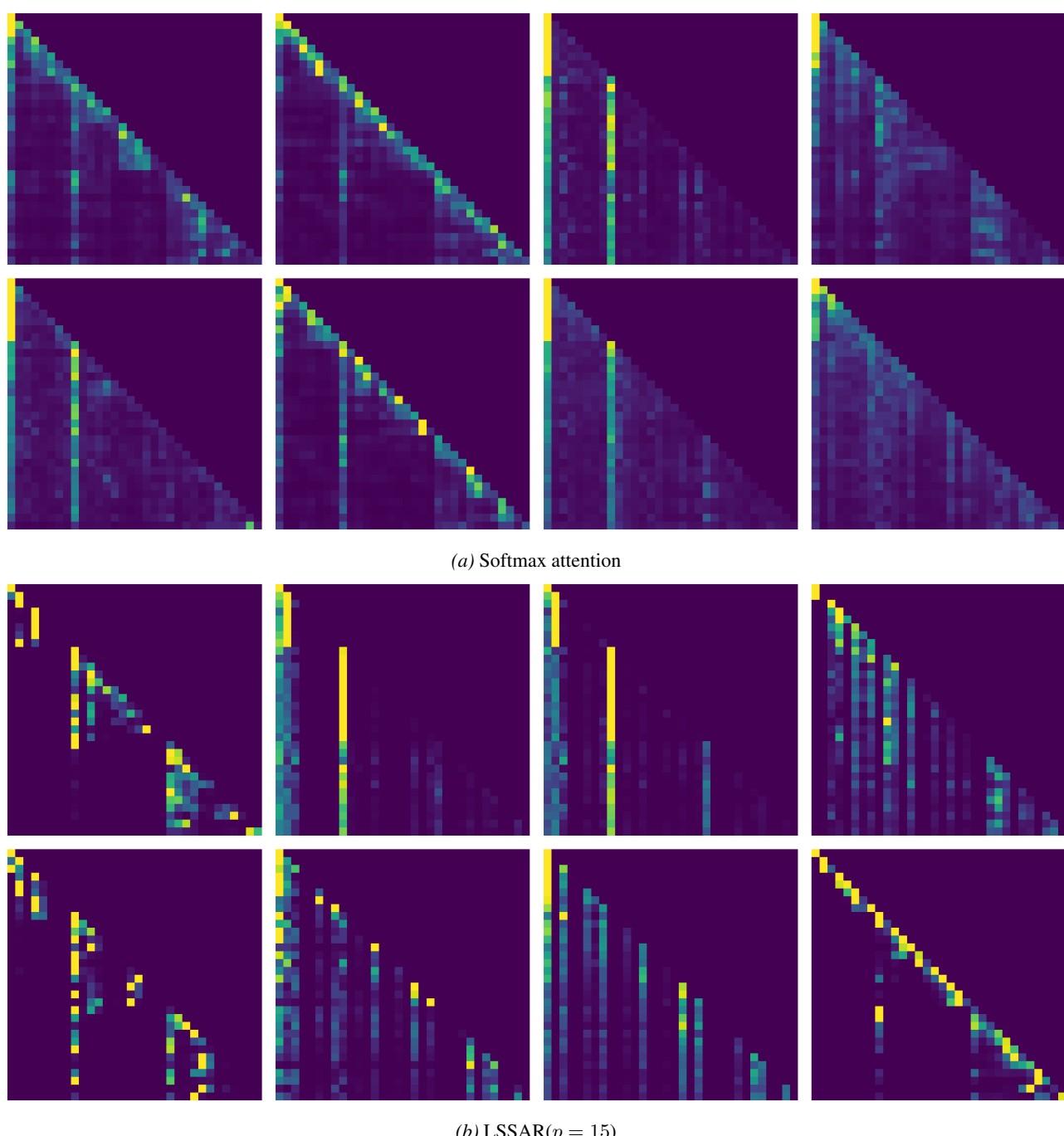

*(a)* Softmax attention

*(b)* LSSAR($p = 15$)

*Figure A.4.* Comparison of attention maps from the last layer of GPT-2-45M, showing standard Softmax attention (above) versus LSSAR with $p = 15$ (below). Each panel displays the 8 attention heads in a 4x2 grid. For visualisation purposes, attention scores are clamped to the range $[0, 0.5]$.

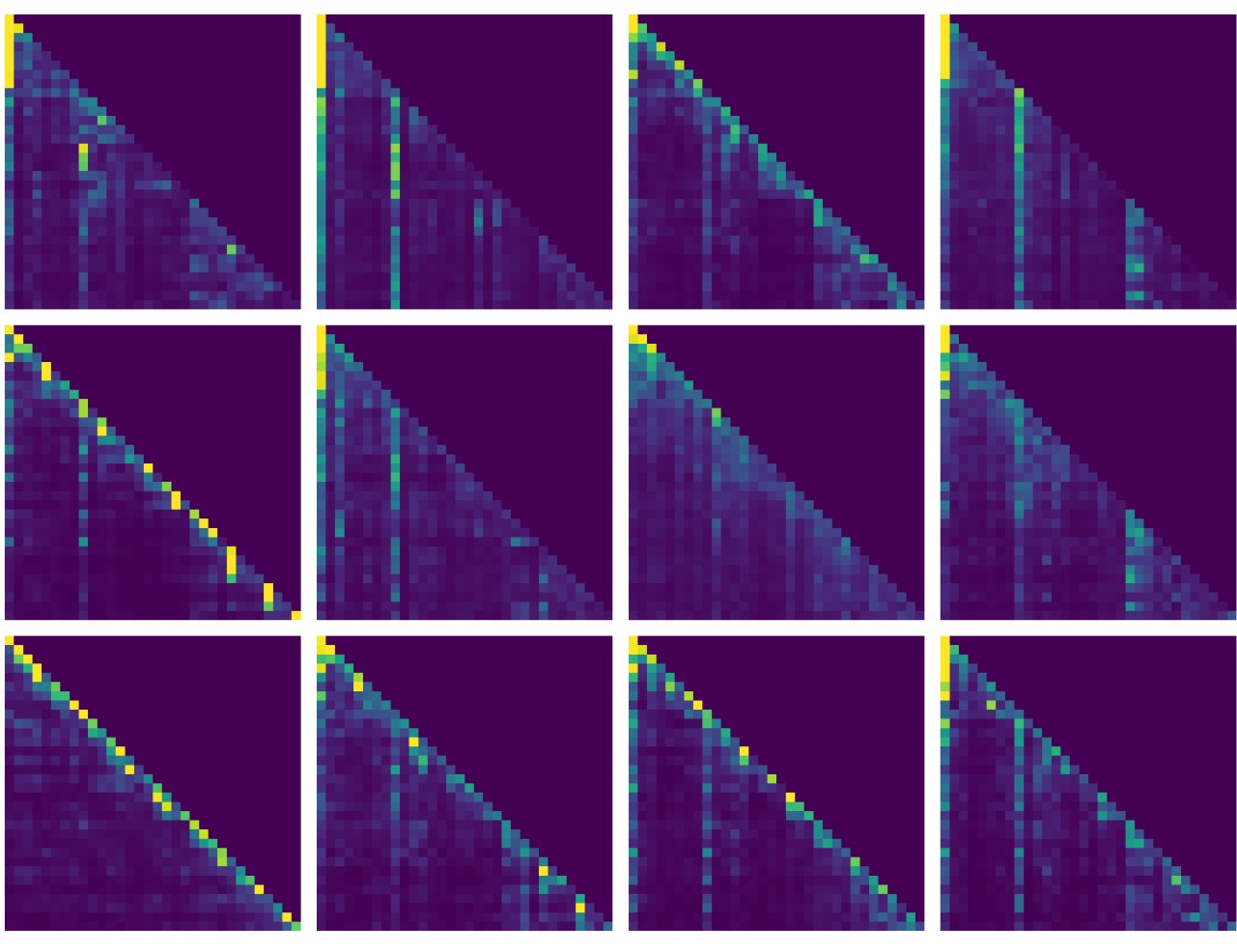

*(a)* Softmax attention

*Figure A.5.* Comparison of attention maps from the last layer of GPT-2-124M, (continued on next page).

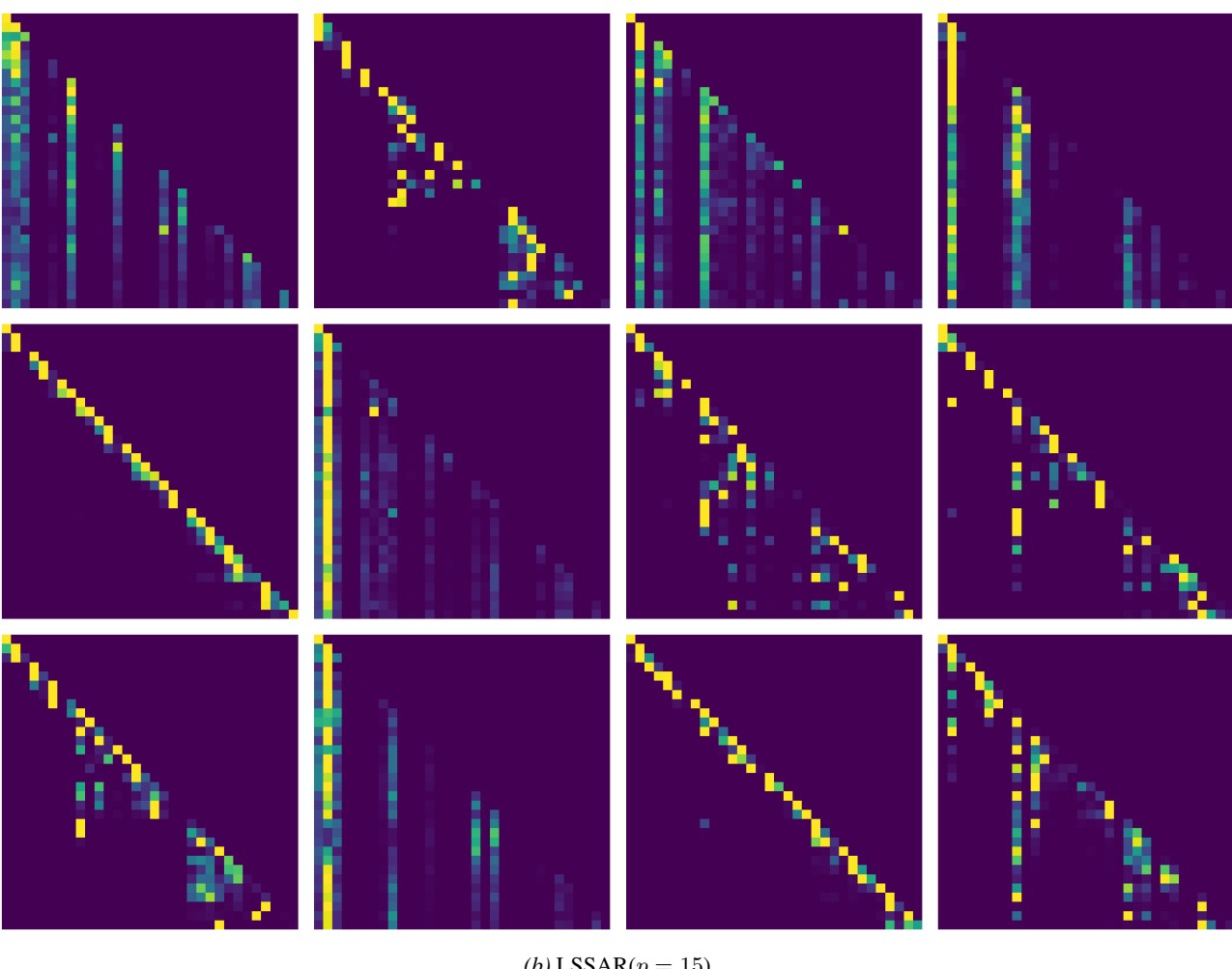

*(b)* LSSAR($p = 15$)

*Figure A.5.* Comparison of attention maps from the last layer of GPT-2-124M, showing standard Softmax attention (above) versus LSSAR with $p = 15$ (below). Each panel displays the 12 attention heads in a 4x3 grid. For visualisation purposes, attention scores are clamped to the range [0, 0.5].

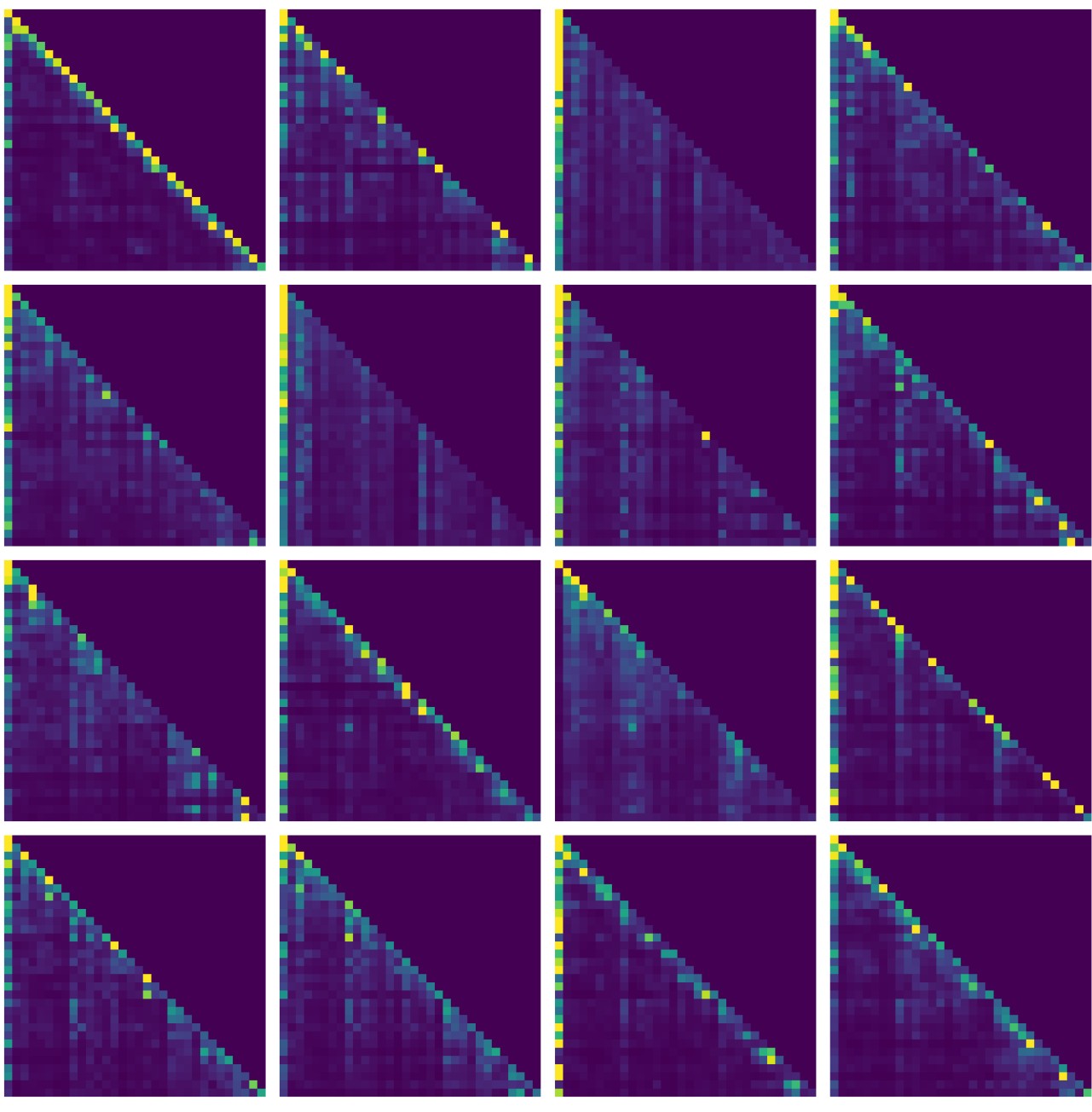

*(a)* Softmax attention

*Figure A.6.* Comparison of attention maps from the last layer of GPT-2-355M (continued on next page).

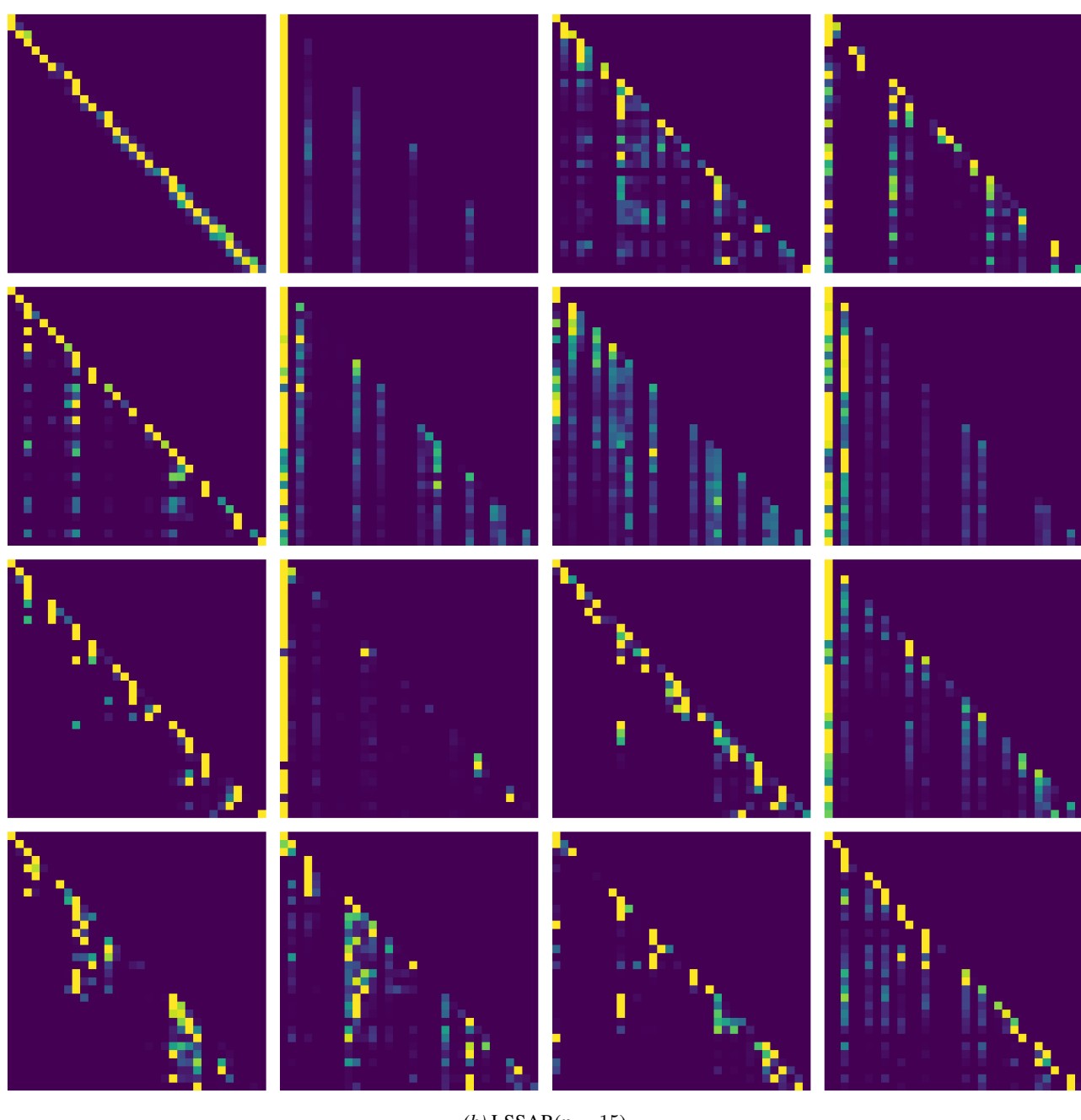

*(b)* LSSAR($p = 15$)

*Figure A.6.* Comparison of attention maps from the last layer of GPT-2-355M, showing standard Softmax attention (above) versus LSSAR with $p = 15$ (below). Each panel displays the 16 attention heads in a 4x4 grid. For visualisation purposes, attention scores are clamped to the range [0, 0.5].

local semantic cues. The results, presented in Tab. A.2, reveal a striking contrast between models trained on FineWeb-10B and FineWeb-Edu.

*Table A.2.* Zero-shot performance on LAMBADA for models trained on different datasets. Note that the 124M Softmax model trained on FineWeb-Edu exhibits severe collapse.

| Model Size (Dataset) | Attention | Accuracy ($\uparrow$) | Perplexity ($\downarrow$) |
|---|---|---|---|
| 45M (FineWeb-Edu) | Softmax | 5.86 | 1397.36 |
| | LSSAR | **18.57** | **361.96** |
| 124M (FineWeb) | Softmax | **31.51** | **45.12** |
| | LSSAR | 30.52 | 48.27 |
| 124M (FineWeb-Edu) | Softmax | 8.79 | 10720.58 |
| | LSSAR | **30.37** | **48.35** |
| 355M (FineWeb-Edu) | Softmax | 12.40 | 263.69 |
| | LSSAR | **35.47** | **33.27** |

The results in Tab. A.2 reveal a critical vulnerability in standard Softmax attention. When comparing the 124M models, the version trained on the standard FineWeb dataset achieves a respectable accuracy of 31.51%. However, the same architecture trained on the rigorously filtered FineWeb-Edu dataset suffers a catastrophic collapse, with accuracy dropping to 8.79% and perplexity exploding to over 10,000. This indicates that while "textbook-quality" data (FineWeb-Edu) may improve performance on scientific benchmarks, it induces a severe bias in Softmax attention that destroys its ability to handle the narrative, long-range dependencies required by LAMBADA. The standard Softmax mechanism appears unable to generalise from the highly coherent training distribution to out-of-distribution contexts.

In stark contrast, LSSAR demonstrates exceptional stability and invariance to data distribution shifts. For the 124M model, LSSAR achieves similar performance (around 30.4% accuracy) regardless of whether it was trained on FineWeb or FineWeb-Edu, completely avoiding the collapse observed with Softmax. Furthermore, the 355M LSSAR model effectively leverages the increased capacity provided by the FineWeb-Edu data, achieving the best overall performance. This confirms that LSSAR's re-weighting mechanism successfully counteracts the overfitting risks associated with highly filtered data, ensuring that the model maintains precise, long-range attention capabilities essential for robust reasoning and extrapolation.

### B.7.1. IMPLICATIONS FOR REASONING MODELS

Recent advancements in reasoning models, such as DeepSeek-R1 (Guo et al., 2025), have demonstrated that reinforcing a "Chain of Thought" (CoT) process can lead to emergent thinking capabilities. These models rely on generating extensive intermediate reasoning steps to solve complex problems. The properties of LSSAR discussed in this paper suggest it could be particularly advantageous for such architectures.

The core mechanism of reasoning models involves extending the sequence length to accommodate profound logical derivations. Standard Softmax attention suffers from attention smoothing as sequence length increases, which dilutes the model's focus and limits the effective depth of reasoning. LSSAR, with its entropy-invariant scaling and sharpening stage, maintains distinct attention peaks regardless of sequence length. This theoretically enables models to sustain coherent thinking processes over indefinitely long sequences without losing track of critical premises established early in the chain.

Furthermore, logical deduction is inherently precise. A conclusion often hinges on a specific, discrete piece of information rather than a diffuse context. The re-weighting mechanism in LSSAR forces the model to make decisive attention allocations, effectively filtering out noise and ensuring that the reasoning process attends to the exact tokens required for the next logical step. As evidenced by the Passkey Retrieval and LAMBADA results on FineWeb-Edu, this precision is maintained even when the model is trained on highly coherent data that typically induces smoothing in Softmax models.

Finally, reasoning models, which often employ RoPE, are susceptible to the attention sink phenomenon, where a significant portion of attention capacity is wasted on the initial tokens (Xiao et al., 2023). LSSAR mitigates this issue (Appendix B.5), promoting a more balanced distribution where all attention heads are utilised for semantic processing. For reasoning-heavy models, this efficient utilisation of attention capacity is crucial for capturing the complex, multi-faceted relationships inherent in deep logical tasks.

Building upon the foundations of R1, the recently released DeepSeek-V3.2 (Liu et al., 2025) introduces DeepSeek Sparse Attention (DSA) and advanced Group Relative Policy Optimisation (GRPO) to further push the frontiers of reasoning. LSSAR offers complementary advantages to these architectural innovations, particularly in enhancing the efficacy of sparse attention mechanisms. While DSA improves computational efficiency by selecting a top-$k$ subset of tokens via a lightning indexer, the standard Softmax applied within this retrieved subset remains liable to smooth attention scores, particularly as $k$ increases or the context expands. Replacing the internal Softmax of DSA with LSSAR would provide a crucial sharpening effect within the sparse window, ensuring that the model attends to the most relevant tokens among the candidates, thereby maximising the information density of the sparse selection.

Furthermore, LSSAR contributes to the stabilisation of large-scale reinforcement learning. DeepSeek-V3.2 employs complex stabilisation techniques for GRPO, such as unbiased KL estimation, to counteract the instability often observed with exponential activations. LSSAR relies on the Softplus activation, which exhibits linear asymptotic growth, as opposed to the exponential growth of Softmax. This fundamental change in gradient dynamics offers improved numerical stability, potentially mitigating the policy gradient variance inherent in large-scale RL training and offering a more robust optimisation landscape for methods like GRPO.

Crucially, LSSAR addresses the token efficiency gap observed between open and proprietary models. As highlighted in Liu et al. (2025), the DeepSeek-V3.2-Speciale model often requires substantially longer reasoning trajectories to match the performance of Gemini-3.0-Pro, a phenomenon attributed to "redundant self-verification" and lower "intelligence density". We posit that this redundancy stems from the uncertainty induced by attention smoothing in Softmax, which necessitates verbose verification loops. By enforcing a more decisive attention distribution through its re-weighting mechanism, LSSAR theoretically enables the model to retrieve premises with higher confidence, thereby compacting the reasoning chain. This improvement directly alleviates the context bottleneck in agentic workflows; by reducing trajectory bloat and leveraging LSSAR's proven length extrapolation capabilities (Appendix B.6), future agents could retain full interaction histories without resorting to destructive context management strategies like "Discard-all".

## B.8. Force Predictions for Other Planets

This section provides visualisations of the gravitational force vector predictions for planetary orbits in the solar system, complementing the Earth orbit analysis presented in Section 3.4. These figures examine whether the qualitative force-vector structure observed for Earth is consistent across planets with varying orbital characteristics.

Across all seven planets shown in Fig. A.7, LSSAR produces force vectors that generally point toward the Sun (the gravitational centre). In contrast, the standard GPT model with Softmax attention generates force predictions that lack coherent radial structure. These visualisations provide qualitative support for the symbolic regression results in Tab. 3: the inverse-square functional form recovered by LSSAR is consistent across multiple planetary orbits rather than being an artefact of fitting to a single trajectory.

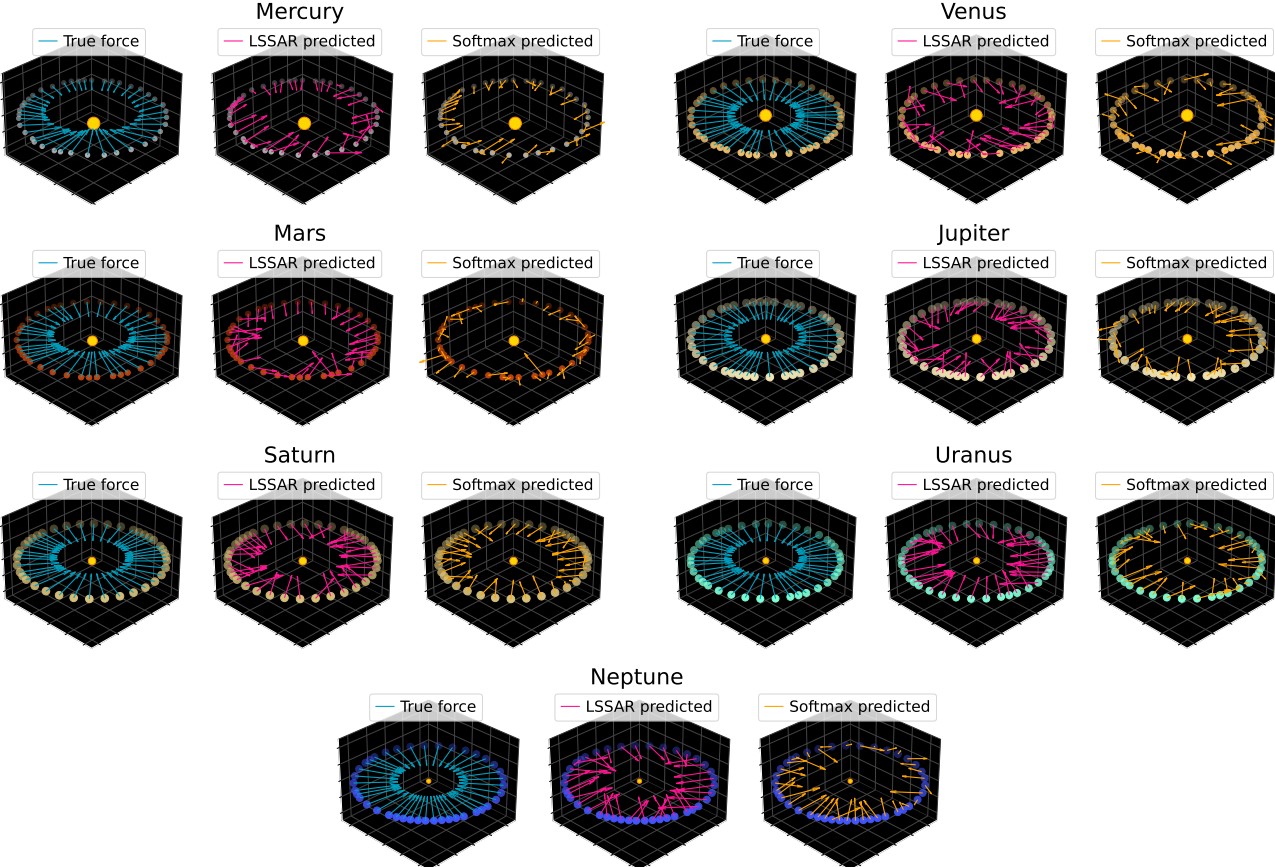

*Figure A.7.* Gravitational force vector predictions for planetary orbits in the solar system (excluding Earth, shown in Fig. 1). For each planet: left panel shows true Newtonian forces, middle panel shows GPT (LSSAR) predictions, and right panel shows GPT (Softmax) predictions. LSSAR generally produces force vectors that point toward the Sun.

