# OpenReview forum: "Softplus Attention with Re-weighting Boosts Length Extrapolation in Large Language Models"
_ICML.cc/2026/Conference — ICML 2026 regular_

### Official Review · Reviewer_QtTe · 2026-03-12

**Soundness:** 2
**Presentation:** 2
**Significance:** 3
**Originality:** 3
**Overall Recommendation:** 5
**Confidence:** 4

**Summary:**

This paper considers a replacement for the traditional attention, based on using Softplus, a scale factor based on input length, and a reweighting to sharpen the attention activation distribution. The authors consider ablations on each component of their proposed method LSSAR to validate the importance of each. They find that their method generally outperforms other baselines on a selection of typical language modelling benchmarks, sequence length extrapolation, and on a task measuring the ability of a model to develop intuition for Newton’s law of universal gravitation.

**Compliance With Llm Reviewing Policy:**

Affirmed.

**Final Justification:**

The rebuttal address all the issues I raised. They require modifications but the authors committed to making those changes in their rebuttal, addressing all my concerns. I think that beyond that, as someone working on LLM, it's always interesting to see work go beyond vanilla transformers, so this work lays a foundation for future research.

**Key Questions For Authors:**

**Questions**
1. p4: “transform raw scores by applying a uniform mapping intensity across the entire input domain” what do you mean by “uniform mapping intensity”? Since you italicised it I assume you want to draw attention to that, but I am not sure what you mean. If you meant that it doesn’t collapse values to 0 or any other value in particular as the end of the sentence indicates, I believe injective might be the suitable word here (and even stronger, invertible). Or you could just keep it simply to “each token has non-zero contribution unlike e.g. with ReLU”.
2. p4: eq 6 discussion: in the argument overall, why are you taking the difference of two elements? I believe your argument comes from just looking at the limit of $x_i$, which will be either 0 or $1/j$ where $j$ is the number of elements equal to $x_m$.
3. p5: how many tokens did you use to extrapolate sequence length? We don’t typically 0-shot longer sequence lengths when training LLMs. It’s a fair research question to ask about 0-shot extrapolation, but I suspect a large part of readers of your work will wonder how this can be of benefit in production, where training over long sequences happens at some point (see e.g. DeepSeek tech reports since you mentioned them in the appendix). If you decide to run this experiment, make sure to visualise the frequency of different sequence lengths in your dataset to ensure that there are enough e.g. 8k sequences to train on.


**Suggestions**
1. p3: In the main body, you discuss the softplus and sigmoid functions. Because your contribution depends on them and you discuss their asymptotic behaviours, readers would benefit from you writing their equations in subsection 2.1.2, like you wrote softmax’s. Perhaps also for ReLU$^p$ later to maximise legibility.
2. Tab1: I’d recommend making it clearer that it’s not just the first row (“Re-weighted”) of sections 2 and 3 of the tables that uses $p=3$ and 15, respectively.


**Typos**
1. p3: “It is often assumed that its non-negative outputs are essential for good performance. However, this assumption is questionable, as replacing Softmax with other non-negative activation functions often leads to a decline in model performance” the logical statement of the first sentence is essentially that non-negative outputs are required for good performance, and the previous works in the second sentence support the idea that it is not sufficient. So I’d rethink “this assumption is questionable”, because the two sentences are not contradictory. Note that I’m not saying you can’t question the first claim generally, it’s just that previous works cited don't contradict it.
2. p3: “Our results (see Tab. A4 in Appendix B.1.1) indicate that the crucial component of Softmax is not its positivity but the normalisation it performs.“ well it’s not the normalisation, it’s the conjunction of the normalisation and the non-linear transformation $e^x$.
3. p4: right above equation 6, the limit of $p$ to $\infty$ is subscripted.

**Limitations:**

No, see weaknesses.

**Strengths And Weaknesses:**

**Strengths**
1. I appreciate reporting the time and memory costs. The authors also pointed accurately to the fact that several operations in attention are memory bound. So are norms. The question of how well this can be addressed in practice remains open, but it’s fine; most papers in conference don’t propose optimised kernels for new methods. However, see weakness #5.
2. I think research into understanding the limits of vanilla attention and studying possible replacements is always of great benefit to the community, and could be of interest to readers working on applications of foundation models. In particular, I appreciate the consideration of biases encoded for tasks relevant to world modelling.
3. I believe the experiments are certainly not enough to convince developers of foundation models to swap to LSSAR, more about this below in my ask to discuss limitations. However, research is iterative, and I believe this paper is promising and can open the door to further work in that direction.
4. The paper is well-written and easy to follow. See suggestion on trying to have the formulas of all components of your algorithm in the main body of your paper to further improve its legibility.

**Weaknesses**

Generally, I think the paper is interesting. However, I’d like to have the following weaknesses addressed, and my questions answered, before I consider raising my score.
1. Missing important reference: *Scalable-Softmax Is Superior for Attention* paper is from over a year ago, but is not cited here; yet it makes scaling softmax on input size not novel, and in that paper, they have an additional learnable parameter, so it’d be interesting to know your work compares.
2. A lot of experimental details are missing (hyperparameters, schedules, …).
3. I believe claims of Newton’s law of gravity being fit by LSSAR need to be toned down. You did better than some production models, yes, and with a much smaller model presumably, so it’s genuinely interesting! However, the model is still significantly off predicting $G \cdot m_{sun}$; so you can’t claim either that the law was predicted given the massive error in the constant. Also it’s worth explaining in detail how the task was set up (prompts, etc), and possibly give something like the average error on the prediction of the force of the different baselines (at the very least, yours).
4. Limitations need to be discussed in the main body of the paper. As I mentioned, the number of tokens and model size are tiny and readers familiar with training foundation models are used to seeing trends reverse past a certain size of data and/or models when experimenting with LLMs. This per se does not mean your paper should be rejected because I understand that this research can still be of interest to readers with more computational power or who would use your work as a foundation for theirs. However, considering the amount of papers claiming an improvement over SotA and the cost of experimenting with foundation models, it’s particularly important to be clear about such limitations. For example, because the training happens over tiny amounts of steps, it is unclear how stable LSSAR is compared to traditional attention, how the curse of depth affects it (softplus does saturate and so does its derivative), etc.
5. Similarly, the results and potential discussion and limitations of computational and memory costs need to have a bigger place in the main body. As it is, I think the text in the main body and appendix is a tad too optimistic; it’d be more balanced to say that 1. It’s left to future work and 2. There are reasons to think it can be significantly improved (followed by your analysis). See more details about weaknesses k#3.

**More details about weaknesses**
1. eq 6 is subtly incorrect in the edge case where more than one of the $x_k$ is equal to $x_m$.
2. p7: “while trillion-parameter LLMs fail” careful: do you have a source for the size? If so you need to indicate it, otherwise… because this is a scientific paper, you cannot claim that, “citation required”.
3.  “Theoretical analysis provided in Appendix B.2 confirms that LSSAR is compatible with optimized implementations such as FlashAttention-3 and can achieve efficiency comparable to standard attention.” is the only reference in the main body, and glosses completely over the fact that as it is, LSSAR is not competitive at all. I appreciate the discussion around improving it, but it’s not clear you’ll 100% close the gap as this seems to be suggesting. Again, this is a scientific paper, so it’s fine to say “we believe that”, “promising”, etc., but not to claim that it is “confirm[ed]” by the discussion I see in the appendix, because your theoretical analysis of the fused operation is nowhere near being detailed enough to support such a claim. And, again, experimenting with foundation models is expensive, a lot of papers claim to propose a SotA method, and yours is almost twice as slow at inference with +80% memory usage, but the reader is told with the current writing that this isn’t a concern at all and that they’ll trivially write the optimised kernels for LSSAR that will erase this disadvantage. At least that’s my read, and as you can guess, it could affect the credibility (and hence impact) of your work. If it were trivial, then you should write the kernel and show it. What’s more, the number of operations after scalar multiplication of QK is significantly higher at any rate which comes with memory AND computational costs (softmax is just an exponential, you need to compute on top of it scalar products, differences, a log, the ReLU and exponentiation to p before normalising, and I agree it’s very optimisable with online tricks as in FlashAttention but it’s still fundamentally more computations), and it is not trivial how you will avoid the numerical instability that is taken care of by $e^(x_i - \text{max}_k x_k)$ in traditional attention, and for which FlashAttention provides a memory-efficient implementation.

---

> ### Author Rebuttal · Authors · 2026-03-29
>
> We thank the reviewer for the thorough review.
>
> **W1: Missing Scalable-Softmax reference**
>
> We agree that scaling Softmax attention input by $\log(L)$ (i.e., $QK^T \cdot \log(L)$) has been previously studied, as discussed in Sect. 2.1.2 (lines 126-133). The reviewer's additional point is that Scalable-Softmax (SSmax) introduces a learnable parameter $s$, making it an important direct baseline. We therefore trained SSmax under the same settings (initialising $s = 0.43$). The results ([validation loss](https://anonymous.4open.science/r/ssmax-2544/vallloss.pdf), [passkey](https://anonymous.4open.science/r/ssmax-2544/passkey.pdf)) show that SSmax improves over vanilla Softmax but remains far behind LSSAR on length extrapolation. We will add the citation and these results.
>
> **W2: Missing experimental details**
>
> Please refer to our Response 4 to Reviewer dqyZ for the full experimental settings.
>
> **W3: Newton's law claims need toning down**
>
> We agree. The fitted constant is substantially inaccurate ($27.02$ vs $4\pi^2 \approx 39.48$, about 32% error), so we should not claim that LSSAR fully recovers Newton's law. We will narrow the claim to recovering the inverse-square functional dependence, i.e., a form proportional to $m_1 / r^2$, rather than the exact physical constant. We will also describe the task setup more clearly (trajectory data, symbolic-regression / prompting protocol, and fitting procedure) and report a quantitative force-prediction error metric.
>
> This narrower result remains interesting. In Vafa et al. (2025), GPT-2 trained on 10M simulated orbital-trajectory sequences and the frontier LLMs they evaluate (o3, Claude 4 Sonnet, and Gemini 2.5 Pro) do not recover the inverse-square relation. In contrast, replacing Softmax attention with LSSAR yields an interpretable inverse-square form rather than the expressions produced by our Softmax baseline that failed to match the physical form (Table 3). We will attribute the trillion-parameter wording explicitly to Vafa et al. (2025).
>
> **W4: Limitations in main body**
>
> Agreed. We will discuss these limitations explicitly in the main body: (a) evidence is currently limited to 45M-355M models, 10B-100B training tokens, and relatively short training; (b) whether the observed gains persist or reverse at larger scales remains unknown; and (c) LSSAR's stability relative to standard Softmax attention, especially in very deep networks where Softplus and its derivative (Sigmoid) can saturate, remains untested. We will present these as open questions.
>
> **W5: Computational cost / FlashAttention claims**
>
> We concede this fully. Current implementation is ~2x slower with ~80% more memory. We will rewrite "confirms compatibility" as "preliminary feasibility analysis suggesting potential compatibility," state that no optimised kernel exists, acknowledge the extra operations (softplus, log, $\mathrm{ReLU}^p$, extra norm), note that it remains unclear whether FlashAttention's $e^{x-\max}$ trick transfers to LSSAR, and frame kernel fusion as future engineering work rather than a resolved result.
>
> **Detail 1 (Eq. 6 edge case)**: Correct, when $j$ values tie at $x_m$, the limit is $1/j$ not $1$. Will fix and state the tied case explicitly.
>
> **Detail 2 ("Trillion-parameter" citation)**: Yes, the source is Vafa et al., cited at the start of the relevant paragraph in W3. We will make this attribution even more explicit in revision.
>
> **Detail 3 (FlashAttention)**: Addressed in W5.
>
> **Q1 ("Uniform mapping intensity")**: The reviewer is right that this wording is unclear. We simply mean that each token retains a non-zero contribution, unlike mappings such as ReLU that can collapse some inputs to zero. We will rewrite the sentence in that simpler form.
>
> **Q2 (Eq. 6 argument)**: The reviewer is right that our exposition there is unnecessarily indirect. The intended argument is a per-element limit analysis of $x_i^p / \sum_k x_k^p$ as $p \to \infty$, not a difference-between-two-elements argument. If $x_i < x_m$, then $(x_i / x_m)^p \to 0$; if $j$ elements attain the maximum $x_m$, each converges to $1/j$. We will rewrite Eq. 6 accordingly and state the tied-maximum case explicitly.
>
> **Q3 (Zero-shot extrapolation)**: Our models are trained with sequence length 1024 and evaluated zero-shot at 2K, 4K, 8K, and 16K. We agree this does not match the standard production pipeline, where continued long-context training is usually introduced. Our intent is narrower: to isolate intrinsic extrapolation capability rather than claim a production-ready long-context recipe. If we add continued long-context training, we will report the sequence-length distribution to verify that sufficiently many long sequences (e.g., 8K) are present.
>
> **Suggestions**: (1) Will add Softplus, Sigmoid, $\mathrm{ReLU}^p$ equations in main body. (2) Will clarify Table 1 formatting for $p=3$ vs $p=15$.
>
> **Typos**: Accepted and will be corrected.
>
> We would be happy to clarify any remaining questions from the reviewer.

---

> > ### Author Rebuttal · Reviewer_QtTe · 2026-04-02
> >
> > > Detail 2 ("Trillion-parameter" citation): Yes, the source is Vafa et al., cited at the start of the relevant paragraph in W3. We will make this attribution even more explicit in revision
> >
> > What I said still applies to them, they're speculating unless there's been announcements for the specific named models. I recommend saying something along the lines of "assumed to be significantly larger" to remain scientifically accurate.
> >
> > __________________________________________________________________________________________________
> >
> > Thank you for your thorough rebuttal and answering all my questions. I am pretty happy with your replies and I trust you with making all the promised changes, in light of you agreeing with the importance of making those modications to your draft.
> >
> > In anticipation of that, I raise my score to an accept (5).

---

> > > ### Author Response · Authors · 2026-04-03
> > >
> > > We sincerely thank the reviewer for the careful follow-up and for raising the score. We appreciate the trust placed in us to make the promised revisions.
> > >
> > > We also appreciate the final clarification regarding the "trillion-parameter" wording. We agree that, to remain scientifically precise, we should avoid stating specific scales unless they are explicitly announced for the named models. In the revision, we will therefore use a more careful phrasing, while keeping the attribution to Vafa et al. (2025) explicit.
> > >
> > > Thank you again for the constructive review. We will make these changes in the revision.

---

### Official Review · Reviewer_u2BX · 2026-03-13

**Soundness:** 3
**Presentation:** 3
**Significance:** 3
**Originality:** 3
**Overall Recommendation:** 5
**Confidence:** 3

**Summary:**

The paper proposes LSSAR, a two stage attention mechanism designed as a direct replacement to standard softmax attention. The method consists of first replacing the softmax with the softplus followed by $\ell_1$ normalization and second a sharping stage which utilizes ReLU$^p$ to amplify the large attention score while suppressing the small one.  Experiments on GPT-2 style models trained on FineWeb-10B demonstrate improved validation loss, improved performance on passkey retrieval tasks, competitive on several NLP benchmarks, and a symbolic regression show the model can recover Newton’s inverse square gravitational law.

**Compliance With Llm Reviewing Policy:**

Affirmed.

**Final Justification:**

The authors addressed my concerns, I provided a positive score, so I will keep the score.

**Key Questions For Authors:**

1. How do we choose $p$?

2. Why does Softmax suffer from “numerical instability” and not Softplus?

**Limitations:**

Yes

**Strengths And Weaknesses:**

Strength:

Soundness: The paper provides a clear mathematical formulation of attention, enabling principled modification of normalization function. In order to resolve the dispersion effect of the normalization stage, the sharping stage is utilized to amplify large attention scores. These are reasonable design choices.

Presentation: The paper is clearly written with explanations of each design choice to resolve particular problems. It is well organized. Results are presented in a manner that is easy to understand.

Significance: Design a model that handles long sequence length is a very important problem for various applications from language to images and video generation. The proposal can be used as a drop-in replacement for standard attention and is useful for its practical relevance.

Originality: The novelty relies on the combination of using the softplus and ReLU$^p$ in replacement to softmax attention.

Weakness:

Soundness: The choice of scaling $\log(d)\log(L)$ is not clearly motivated.

Significance: Performance of the model drops significantly as the sequence length increases, Fig 2. Thus $p$ should be a function of input length. The inevitability of dispersion for a fixed choice of $p$ will occur at large sequence length [2,3].

All experiments are conducted on small scale models making it difficult to access real world impact with modern models.

The proposed method suffers the quadratic computation as to the softmax attention which limits its capability on models that process long sequences.

Originality: The overall design of amplifying the attention score after the normalization is not new. In particular [1] uses a similar approach to amplify the score after a linear attention operation.


[1] Han, D., Pan, X., Han, Y., Song, S., and Huang, G. Flatten transformer: Vision transformer using focused linear attention. ICCV, 2023.

[2] Nhat Thanh, T., Fanghui, X., Shuai, Z., Jiancheng, L., Yunling Z., Yingyong Q., and Jack, X. “SEMA: a scalable and efficient mamba like attention via token localization and averaging,”
arXiv:2506.08297, 2025.

[3] Velickovic, P., Perivolaropoulos, C., Barbero, F., and Pascanu, R. Softmax is not enough (for sharp size generalisation). arXiv:2410.01104, 2024.

---

> ### Author Rebuttal · Authors · 2026-03-29
>
> We thank the reviewer for the feedback.
>
> **1. $\log(d)\cdot\log(L)$ scaling not clearly motivated**
>
> Detailed in Sect. 2.1.2, $\log(d) \cdot \log(N)$ serves two distinct purposes:
> (1) **$\log(d)$**: Using $\ell_2$-normalised $Q$ and $K$ bounds their product to $[-1, 1]$. In high-dimensional spaces, random vectors are nearly orthogonal, so the $QK^T$ dot product concentrates tightly near zero with very small variance. Passing this compressed range through Softplus yields minimal signal separation. Scaling by $\log(d)$ stretches this distribution, restoring dynamic range and distinguishing power between tokens before the nonlinear mapping.
> (2) **$\log(N)$**: Following Su (2021), this maintains entropy invariance as sequence length grows. Without $\log(N)$, attention distributions flatten drastically over long contexts. Scaling each row $i$ by $\log(i)$ preserves effective temperature and drives extrapolation.
> We will highlight these existing derivations.
>
> **2. $p$ should be function of input length; dispersion inevitable**
>
> We agree and evaluated a position-dependent adaptive $p_i$ for query row $i$:
> $p_i = 15$ if $i \le L_{\text{train}}$, else $p_i = 15 \cdot (1 + \alpha(\sqrt{i / L_{\text{train}}} - 1))$
> where $L_{\text{train}} = 1024$ and $\alpha$ controls extrapolation strength.
>
> Testing $\alpha \in \{0.1, \dots, 0.5\}$ (see [validation loss](https://anonymous.4open.science/r/dyp-6C75/valloss.pdf), [passkey accuracy](https://anonymous.4open.science/r/dyp-6C75/passkey_accuracy.pdf)), validation loss at extreme contexts improves (e.g., at 16× extrapolation, $\alpha=0.1$ reduces loss from 3.45 to 3.31). However, passkey retrieval degrades similarly to fixed $p=15$, confirming dispersion while our LSSAR experiments still demonstrate considerable extrapolation benefit. We will include this adaptive analysis.
>
> **3. Small-scale experiments**
>
> We acknowledge this limitation. Due to academic cluster constraints, exhaustive 1B+ baseline comparisons from scratch are intractable.
>
> However, evaluating LSSAR and Softmax at 45M and 355M scales (App. B.4) shows consistent relative improvements across a 7.9× range. We will state larger-scale validation as an important next step in limitations.
>
> **4. Quadratic computation**
>
> Yes, LSSAR retains $O(L^2)$ complexity. As emphasised in Sect. 5 (lines 405–420), our contribution is a mechanism-level replacement within the exact attention paradigm, not a sub-quadratic approximation.
>
> **5. FlattenTransformer uses similar amplification**
>
> Important architectural differences exist:
> (1) FlattenTransformer applies a norm-preserving feature mapping $\phi_p(x) \propto \text{ReLU}(x)^p$ to individual $Q$ and $K$ vectors. While this preserves individual vector norms ($\|x\| = \|\phi_p(x)\|$), it merely alters feature directions before linear attention approximation, thus losing exact pairwise scores.
> (2) LSSAR applies shifted power function $\text{ReLU}(\mathbf{A}\otimes \mathbf{N}-\mathbf{O})^p$ directly to exact pairwise dot-product scores before normalisation (Sect. 2.2.2).
>
> Our core contribution is an exact-attention replacement that improves upon Softmax across length extrapolation, passkey, downstream tasks, and physical understanding in our experiments. We will cite FlattenTransformer while clarifying these distinct computational contexts.
>
> **6. How to choose $p$?**
>
> Practically, optimal $p$ depends on training sequence length and architecture. As length increases, a larger $p$ is required. We suggest these provisional heuristics as starting points for tuning: $p \approx 13$ for 1024 length, $p \approx 18$ for 2048, and $p \approx 25$ for 4096.
>
> We will add these practical guidelines to the revision.
>
> **7. Why Softmax suffers numerical instability but not Softplus?**
>
> In our re-weighting experiments, the more direct issue is that Softmax is poorly matched to aggressive sharpening.
>
> Sect. A.6 analyses this directly. For the winner token, the Softmax source gradient decays as $\partial a_i / \partial x_i \propto e^{-x_i}$. Therefore, when $p$ is large, the Softmax-based variant quickly enters an exponential-saturation regime in which the learning signal becomes too weak. This is the behaviour observed in Fig. A3 / Sect. B.1.4, where Softmax with re-weighting degrades sharply at high $p$.
>
> In contrast, with Softplus followed by $\ell_1$ normalisation, the first-stage mapping is milder for large positive logits ($\text{Softplus}(x) \approx x$ for $x \gg 0$), and Sect. A.6 shows a polynomial source-gradient decay ($\partial a_i / \partial x_i \propto x_i^{-2}$) instead of an exponential one. This makes the Softplus-based first stage much easier to optimise under large sharpening factors. We will revise the paper to distinguish this mechanism-specific optimisation issue from the separate finite-precision brittleness of exp-based attention at very large scales.
>
> We would be happy to clarify any remaining questions from the reviewer.

---

> > ### Author Rebuttal · Reviewer_u2BX · 2026-04-03
> >
> > Thank you for the response. I gave a positive score, and I will keep the score.

---

> > > ### Author Response · Authors · 2026-04-05
> > >
> > > We sincerely thank the reviewer for the follow-up and for maintaining the positive score. We are glad that our responses have adequately addressed the concerns.
> > >
> > > We also appreciate the constructive feedback throughout the review process. We will carefully incorporate the promised clarifications and revisions in the final version.

---

### Official Review · Reviewer_jMpi · 2026-03-16

**Soundness:** 2
**Presentation:** 2
**Significance:** 2
**Originality:** 2
**Overall Recommendation:** 4
**Confidence:** 3

**Summary:**

The paper introduces LSSAR, a alternative for the self-attention mechanism that 1) replacing the exponential function in softmax with softplus function, which improves the numerical stability over softmax-based implementation; 2) apply a normalization to sharpen the distribution of attention scores. Numerical experiments show that LSSAR achieves significantly better loss than baseline approaches when extending sequence beyond the length in training, and outperforms classical softmax-based attention in passkey retrival experiment.

**Compliance With Llm Reviewing Policy:**

Affirmed.

**Final Justification:**

The authors' response resolves my concerns on the effectiveness of the method under practical setup. The applicable scope of the method is a bit limited, and i recommend to view it as a borderline paper.

**Key Questions For Authors:**

Please see my comments on weakness.

**Limitations:**

Yes.

**Strengths And Weaknesses:**

**Strengths:**
* The softplus-based attention is easy to optimize given the $\ell_2$ normalization on the pre-attention score.
* The LSSAR method exhibits significantly more stable performance when scaling the sequence beyond the length during training, which makes LSSAR a potential replacement for standard softmax-based attention in long sequence scenario.
* The key retrival experiment further justifies the LSSAR's effectiveness in tasks that requires long-term memory.

**Weaknesses:**
* There has been rich literature that address long context inference issue, e.g. position interpolation [1], which substantially improves the performance in out-of-distribution sequence length. It is suggested to verify the performance of LSSAR when using these techniques, in order to justify the effectiveness of the method under practical setup.
* The experimental setup is not practical. The scale of the experiment is up to 355M, which is far from the practical model size. The number of training tokens is not enough to demonstrate the model's performance as well. For instance, the token number for training 124M model is $4\times1024\times18865 \approx 77M$, which is only $77/(124*20)\approx 3$% Chinchilla law.
* Appendix B.2 mentions that the training precision is pure bfloat16. Is bfloat16 also used for computing softmax? If so, then the baseline's performance is under significant bias and all the numerical results are not convincing anymore.

[1] Chen et al., Extending Context Window of Large Language Models via Positional Interpolation

---

> ### Author Rebuttal · Authors · 2026-03-29
>
> We thank the reviewer for their detailed technical assessment and address each concern below.
>
> **1. Missing comparison with position interpolation (PI)**
>
> We discuss this distinction in Section 1 (lines 78–82) and Section 5 (right, lines 398–410). LSSAR and PI address different components: LSSAR modifies how attention scores are computed and normalised, while PI modifies positional embeddings. Our experiments show that LSSAR achieves 16× zero-shot extrapolation (training on 1024, evaluating at 16K) with nearly constant validation loss using standard RoPE, without any positional modification.
>
> We acknowledge we have not tested LSSAR+PI. The core contribution of our paper is to demonstrate that the attention mechanism itself is a critical and previously under-explored factor in length extrapolation.
>
> **2. Scale too small and insufficient training tokens (~3% Chinchilla)**
>
> We regret that Appendix B.2 lacked clear training detail, causing the token count to appear low. The reviewer's calculation $4 \times 1024 \times 18{,}865 \approx 77\text{M}$ considers only a single GPU's micro-batch. The full computation accounts for 8 GPUs and gradient accumulation of 16 steps: $4 \times 1024 \times 8 \times 16 \times 18{,}865 \approx 9.89\text{B}$ tokens. The actual ratio is $9.89\text{B}/124\text{M} \approx 80$ tokens per parameter. Please refer to our Response 4 to Reviewer dqyZ for the full experimental settings.
>
> Regarding model scale: our main experiments comparing all attention mechanisms are conducted at 124M. As an academic institution, our GPU resources are limited. Our cluster has a total of 32 A100 (80GB) GPUs, and each user can schedule a maximum of 8 at a time via Slurm, with a strict 48-hour time limit per job. Training a model of 1B+ parameters from scratch under these conditions is extremely difficult for us, let alone doing so multiple times to exhaustively compare different attention mechanisms. Under these conditions, obtaining the results for the different $p$ values in Figure A3 took us almost 2 months of intermittent training. Despite these constraints, we additionally trained LSSAR and Softmax at 45M and 355M (Section B.4) to verify scaling trends, showing consistent relative improvement across a 7.9× parameter range.
>
> We will add a limitations section acknowledging that larger-scale validation is an important next step.
>
> **3. bfloat16 training precision — baseline bias?**
>
> We clarify that our baseline experiments were conducted using PyTorch's Automatic Mixed Precision (AMP), rather than strictly pure bfloat16 across all operations. Under AMP, the majority of the network uses `bfloat16`, but PyTorch automatically upcasts its built-in `softmax` function inputs to `float32`. Thus, our Softmax baseline specifically executes its core exponentiation and summation locally in `float32`, which substantially mitigates overflow and precision bottleneck issues.
>
> We are happy to discuss these points further and welcome any additional questions.

---

> > ### Author Rebuttal · Reviewer_jMpi · 2026-04-03
> >
> > Thank authors for the response. I understand that the positional interpolation is a orthogonal method to LSSAR. My main concern is that the paper does not systematically benchmark the performance of combining LSSAR and mainstream long-context training methods such as PI, as there is no evidence that integrating LSSAR into PI will necessarily lead to better performance.

---

> > > ### Author Response · Authors · 2026-04-05
> > >
> > > Thank you for the follow-up. We agree that this is an important question, and we would like to clarify both the scope of the paper and the additional experiments we have now run.
> > >
> > > **1. Scope of the paper**
> > >
> > > Our paper studies the attention mechanism itself: specifically, whether modifying how attention scores are formed and normalised can improve length extrapolation. The motivation is that attention smoothing is a root cause of poor extrapolation, so our focus is on attention computation rather than positional methods. We stated this distinction explicitly in Section 1 (lines 78–82) and Section 5 (lines 398–410), where we explained that LSSAR and PI are orthogonal: LSSAR changes the attention mechanism, whereas PI changes positional encoding.
> > >
> > > For this reason, the original paper did not treat PI as part of the main method under study. Our central claim is that the attention mechanism itself is a critical and previously under-explored factor in length extrapolation.
> > >
> > > **2. Additional LSSAR+PI experiments**
> > >
> > > Given the reviewer's interest in the combined setting, we have now run direct LSSAR+PI experiments at longer context lengths. Starting from our 1024-token LSSAR checkpoint ($p=15$), we performed continued finetuning with PI enabled at 2K, 4K, and 8K context lengths on the same FineWeb 10B dataset.
> > >
> > > The finetuning setup is as follows:
> > > - Context lengths: 2K, 4K, 8K
> > > - Continued finetuning steps: 2000 at each target length
> > > - Learning rate: `6e-5` with 100-step warmup and no further decay
> > > - Total batch size: `524288` tokens
> > >
> > > Validation loss comparison:
> > >
> > > | Target length | Base LSSAR ($p=15$) | LSSAR ($p=15$) + PI |
> > > | --- | --- | --- |
> > > | 2K | 3.1930 | 3.1636 |
> > > | 4K | 3.2291 | 3.1560 |
> > > | 8K | 3.3171 | 3.1638 |
> > >
> > > Passkey accuracy comparison:
> > >
> > > | Target length | Base LSSAR ($p=15$) | LSSAR ($p=15$) + PI |
> > > | --- | --- | --- |
> > > | 2K | 34% | 83% |
> > > | 4K | 15% | 48% |
> > > | 8K | 3% | 21% |
> > >
> > >
> > > These new experiments are intended as a direct check of the reviewer's concern: whether LSSAR continues to provide benefit when combined with a mainstream long-context method such as PI. These results confirm that LSSAR remains highly effective when combined with PI, supporting its practical applicability. At all three target lengths, the combination yields consistent improvements over the original LSSAR baseline in both validation loss and passkey accuracy, with especially large gains on passkey retrieval. We note that the absolute passkey accuracy at 8K (21%) reflects the limited finetuning budget (2000 steps from a 1024-length checkpoint). We expect longer continued training or curriculum-based scaling to further improve these numbers.
> > >
> > > Our primary comparison is therefore between the original LSSAR model and the corresponding LSSAR+PI finetuned model at each target context length, with passkey accuracy as the main long-context metric and validation loss as supporting evidence. We will include these combined LSSAR+PI experimental results in the appendix of the revised paper to verify the method's effectiveness under practical long-context inference setups.

---

### Official Review · Reviewer_dqyZ · 2026-03-18

**Soundness:** 2
**Presentation:** 3
**Significance:** 2
**Originality:** 3
**Overall Recommendation:** 3
**Confidence:** 4

**Summary:**

The paper introduces a two-stage attention mechanism to overcome numerical instability and performance issues in traditional Softmax attention for large language models. It replaces Softmax with a more stable normalization method using Softplus and normalization, along with a dynamic scaling factor, and then applies a sharpening step to better emphasize important tokens while suppressing less relevant ones. This design claims to improve long-context handling, mitigates attention degradation. Experiments show comparisons on long-sequence tasks and benchmarks, and demonstrate the model’s ability to recover Newton’s gravitational law.

**Compliance With Llm Reviewing Policy:**

Affirmed.

**Key Questions For Authors:**

Why are 3 rows of O set to ones? Which theoretical ground to determine the number of rows with value ones?

Besides the example in Appendix B.1.1, could the authors provide more formal or intuitive explanation on the key-role of normalisation?

**Limitations:**

Yes

**Strengths And Weaknesses:**

The paper addresses an important problem in machine learning. The proposed idea is both interesting and well-motivated, and the experimental results are promising. Overall, the paper is clearly written, well-structured, and easy to follow.


It’s interesting to see an example that demonstrate the effect of non-negativity versus  that of normalisation to motivate the work. However, as it is the foundation for the development of the method, it would be more useful for the community to have a formal, comprehensive study on this matter,  rather than developing an ad-hoc approach specifically to enforce an unproven idea.

While using polynomial transformation for reweighing can standout the highest attention value, it would suffers two drawbacks.
- First, it causes the problem of numerical instability when very large p is required (e.g. many high attention values are similar). This contradicts with the purpose of the paper to get rid of Softmax for the same problem.
- The choice of p is not systematic, in the experiments section it’s not clear how the p is selected. This would limit the applicability of the work in practical problems.

The paper should provide more information about the experiment settings for fair comparison.

---

> ### Author Rebuttal · Authors · 2026-03-29
>
> We thank the reviewer for their thoughtful evaluation.
>
> **1. Non-negativity vs normalisation needs formal study**
>
> We acknowledge this is not yet a full formal study. What we provide is a controlled ablation framework: Eq. (3) separates the roles of non-negativity and normalisation, allowing any activation $\phi$ to be substituted. Tables A4–A6 systematically test six activations, suggesting $\ell_1$-norm is the critical component. Additionally, Section 2.2.1 (line 184) provides an analytical perspective on the non-negativity side: any consistent nonlinear mapping whose output contains no zero elements, whether all-positive, all-negative, or mixed, preserves every token's ability to participate in the subsequent $\ell_1$-norm competition. We view this as principled empirical and analytical evidence, though we agree a complete formal treatment would further strengthen the contribution.
>
> **2. Numerical instability of large $p$ in re-weighting**
>
> We appreciate this concern. The reviewer is right that a naive polynomial re-weighting can become numerically unstable when $p$ is large, since in the sharpening stage $z=\text{ReLU}(\text{att}\cdot N-O)$ may exceed 1 and $z^p$ can overflow. However, this is not how our implementation applies the power.
>
> Before exponentiation, we divide each row by its row-wise maximum with `.detach()`: $\hat{z}=z/\max(z)\text{.detach()} \in [0,1]$, and then apply $\hat{z}^p$ followed by $\ell_1$-norm. Thus the power is never applied to values larger than 1. The `.detach()` ensures that this rescaling only stabilises the forward values, without altering the backward gradient path.
>
> This rescaling also does not change the final attention output, because $\ell_1$-norm is scale-invariant: $\text{normalize}((z/m)^p)=\text{normalize}(z^p)$. Therefore, while the reviewer's concern is valid for a naive $z^p$ implementation, our actual implementation avoids this instability by construction. For rows beyond the first 3 (see Q5), $\max(z)>0$ in practice, and Section A.6 suggests the resulting gradients remain well-behaved even for large $p$.
>
> **3. Choice of $p$ is not systematic**
>
> We are transparent about $p$ selection. In Section B.1.4, we swept $p \in \{1, 2, \dots, 15\}$ and evaluated validation loss and extrapolation (Fig. A3). $p=3$ minimises validation loss at training length (1K); for extrapolation, gains continue beyond 13 but become very small. We selected $p=15$ where further increase yields negligible benefit, and tested $p=50, 100$ to confirm degradation at extremes.
>
> This is an empirical selection, not theoretically derived. For practical guidelines on how to choose $p$ based on training length, please refer to our Response 6 to Reviewer u2BX. We will add the practical guidelines to the revision.
>
> **4. Missing experiment settings for fair comparison**
>
> We apologise that Section B.2 lacks sufficient detail. Complete settings: 8 A100 80GB GPUs, GPT-2-124M (12L, 12H, 768d) with RoPE, FineWeb-10B, seq len 1024. Adam ($\beta_1{=}0.9, \beta_2{=}0.95$), lr $6{\times}10^{-4}$, 700-step warmup + cosine decay, weight decay 0.1, grad clip 1.0. Micro-batch 4/GPU $\times$ 1024 $\times$ 8 = 32,768 tokens/step; total batch 524,288, grad accum 16. Training: 18,865 iters $\approx$ 9.89B tokens, val every 1K steps. All baselines share identical setup, only the attention mechanism differs. We will add a hyperparameter table to Section B.2.
>
> **5. Why are 3 rows of $O$ set to ones?**
>
>
> To clarify, $O$ is set to $1$ everywhere except in the first 3 rows, where it is set to $0$. The reason is empirical: very short rows are uniquely fragile. If the offset is also set to $1$ there, row 1 becomes all-zero with certainty after the shift-ReLU step, and rows 2–3 become all-zero with high probability. All-zero attention rows break gradient flow entirely. Through iterative testing, rows 1, then rows 1–2, remained unstable; rows 1–3 was the first stable setting.
>
> Thus, the threshold of 3 rows is a practical stabilisation choice rather than a theoretically derived constant. The theoretical rationale is only that the shortest rows must be protected from collapsing to zero; once this is done, performance becomes largely insensitive to the threshold because rows 4+ retain nonzero entries in practice (Section 2.2.2), consistent with Section A.6.
>
> **6. More formal explanation of normalisation's role**
>
> $\ell_1$-norm was a central design consideration for the re-weighting stage. In Section 2.2.1 (lines 190–202), we analyse how $\ell_1$-norm creates lateral inhibition via gradient coupling: increasing one token's weight forces others to decrease, ensuring suppressed tokens receive gradient feedback and preventing the dead neuron problem. This analysis directly motivated applying $\text{ReLU}^p$ after $\ell_1$-norm rather than on raw scores. Tables A4–A5 confirm that removing $\ell_1$-norm degrades all attention variants significantly.
>
> We are happy to clarify any remaining questions.

---

> > ### Author Rebuttal · Reviewer_dqyZ · 2026-04-03
> >
> > Thank you for the responses. The implementation strategy described for handling large $p$ appears closely related to the well-established stabilization technique used in softmax computations, where the maximum input value is subtracted to improve numerical stability. Therefore, it remains unclear what “numerical instability” the paper is trying to improve from Softmax.
> > Thank you for clarifying the selection of $p$ based on the validation loss. Do you use a held-out test set to report the results or still use the same validation set?
> > While the intuition behind the approach is compelling, the approach remains partly empirical and would benefit from deeper theoretical analysis to substantiate its validity and generality.

---

> > > ### Author Response · Authors · 2026-04-05
> > >
> > > Thank you for this clarification. We realise that our previous response misunderstood the reviewer's original concern. We interpreted the question as asking whether LSSAR's power operation $z^p$ itself causes numerical overflow for large $p$, and accordingly explained our row-wise max rescaling. We now understand the reviewer to be asking a different and more important question: in our large-$p$ re-weighting setting, why does the Softmax-based variant become unstable or degrade, whereas LSSAR remains well-behaved?
> > >
> > > **1. Optimisation dynamics under sharpening**
> > >
> > > App. A.6 analyses the asymptotic gradient behaviour of both Softmax+re-weighting and LSSAR. For the winner token, the Softmax source gradient decays as $\partial a_i / \partial x_i \propto e^{-x_i}$, whereas the Softplus-based first stage decays polynomially as $\partial a_i / \partial x_i \propto x_i^{-2}$. This exponential saturation is the core reason Softmax degrades under sharpening: when $p$ is large, the Softmax-based variant quickly enters a regime where the learning signal becomes too weak, which is precisely the behaviour observed in Fig. A3 and App. B.1.4.
> > >
> > > **2. Two distinct claims about Softmax**
> > >
> > > We agree that our row-wise max rescaling before $z^p$ is analogous in spirit to the subtract-max trick used in Softmax: both are standard forward stabilisation techniques. Our point is therefore not that Softmax is problematic merely because it also requires such a trick. Rather, the distinction we intend is between two different issues: a broader finite-precision brittleness associated with exp-based attention in large-scale settings, and the mechanism-specific optimisation behaviour under aggressive sharpening.
> > >
> > > This ambiguity partly stems from how our paper currently presents these claims together. As stated in the abstract, we identify two separate limitations of Softmax: (i) finite-precision numerical brittleness of the exponential function $e^x$ when scaling to very large models, and (ii) attention score smoothing as token length increases. These are distinct issues. For the specific phenomenon in Fig. A3 / App. B.1.4, the key mechanism is the optimisation mismatch between Softmax and large-$p$ re-weighting (as analysed in Section 1 above), not a forward numerical stability problem. We agree the paper currently conflates these two points and will revise to separate them clearly.
> > >
> > > **3. Selection of $p$**
> > >
> > > We thank the reviewer for raising this point. To answer directly: we did not use a separate held-out test set; $p$ was selected and evaluated on the same validation split. That said, Fig. A3 is intended as a sensitivity study verifying the trend predicted by Eq. 6, rather than a fine-grained model-selection experiment: increasing $p$ sharpens the attention distribution and improves extrapolation, but beyond a certain point the gains plateau and eventually degrade. The performance curve is relatively flat across a broad interval (e.g., $p \in [11, 15]$), so the selected $p$ should be interpreted as an estimate within this plateau rather than a tightly optimised point. This is consistent with our broader empirical observation that the preferred $p$ mainly tracks training sequence length and architecture, rather than requiring fine-grained tuning. We will clarify this protocol in the revision.
> > >
> > > **4. "Partly empirical"**
> > >
> > > We appreciate this observation and agree that further theoretical development, such as formal convergence guarantees, would strengthen the work. We will revise the paper to make this scope clearer.
> > >
> > > At the same time, we would like to clarify that the submission is not purely empirical. App. A already provides several nontrivial analyses, including the structural comparison with Sparsemax/Entmax (A.3), the log-space / thermodynamic interpretation of re-weighting (A.4), and the asymptotic gradient analysis in A.6 showing exponential versus polynomial source-gradient decay under sharpening. The main text also analyses Eq. 6 (convergence to argmax), Sect. 2.1.2 ($\log(d)\cdot\log(N)$ scaling), and Sect. 2.2.1 (gradient coupling / lateral inhibition). We view these as mechanistic analyses that provide analytical grounding for the proposed mechanism, while acknowledging that stronger formal guarantees remain an open direction.

---

### Decision · Program_Chairs · 2026-04-30

**Decision:**

Accept (regular)

**Comment:**

This paper introduces a novel, two-stage attention mechanism designed to replace the standard Softmax operation in large language models. The authors address the numerical instability and performance degradation associated with scaling inference token lengths. The submission presents a compelling and well-executed advancement in attention mechanisms, tackling one of the most critical bottlenecks in current foundational model architectures: robust long-sequence modeling. Overall, the paper provides a relevant contribution to the ICML community.